# Two δ-catenins, plakophilin 4 and p120, promote formation of distinct types of adherens junctions

Indrajyoti Indra[1] , Regina B. Troyanovsky[1] , Farida V. Korobova[2,3] , and Sergey M. Troyanovsky[1,3]

**Classical cadherins are instrumental for connecting cells into tissues by forming adherens junctions (AJs), a structurally diverse class of cell–cell adhesions tailored to specific membrane domains, cell types, and particular functions. The mechanisms that underlie the AJ diversification remain unknown. Here, we show that two δ-catenin family members, p120 and plakophilin 4 (pkp4), which bind the intracellular region of classical cadherins, promote distinct modes of cadherin clustering, thereby contributing to AJ specialization. The mode promoted by p120 is driven by interactions between cadherin-associated protein, α-catenin, and actin filaments. This "canonical" clustering mechanism generates apical and basal AJs that play a major role in overall cell–cell adhesion. The mode promoted by pkp4 is driven by an α-catenin–independent mechanism. It generates lateral AJs, which apparently function in processes other than cell–cell adhesion. Collectively, our findings show that δ-catenins regulate the balance between different AJ assembly pathways, thereby contributing to AJ diversification.**

## Introduction

Adherens junctions (AJs) are cell–cell adhesive structures that interconnect neighboring cells in nearly all multicellular organisms (Honig and Shapiro, 2020; Mège and Ishiyama, 2017; Miller et al., 2013; Troyanovsky, 2023). In addition to their primary role in adhesion, AJs participate in a myriad of signal transduction pathways, including those involved in force sensing (Lin et al., 2023; Mendonsa et al., 2018; Sisto et al., 2021; Yap et al., 2018). The mechanisms underlying the remarkable functional diversity of AJs remain poorly understood.

AJs are highly variable in morphology, localization, and composition of accessory proteins (Franke, 2009; Franke et al., 2009; Takeichi, 2014). The best-characterized AJs, the apical AJs, are located at the apical region of the lateral plasma membrane and can be subdivided into two subtypes: linear AJs (also known as zonula adhaerens) and punctate AJs (also referred to as fascia adhaerens, focal, or radial AJs). These two subtypes can interconvert, producing all possible intermediate forms. A second type, the basal AJs, are morphologically similar to the punctate apical AJs but form at the basal portion of the lateral membrane (Barai et al., 2025; Indra et al., 2013; Kroeger et al., 2024; Pradhan et al., 2023). Unlike the relatively immobile apical AJs, basal AJs, at least in some epithelial cells in culture, exhibit directed upward movement (Hong et al., 2010; Kametani and Takeichi, 2007). Both apical and basal AJs play major roles in intercellular adhesion by interconnecting the actomyosin cytoskeletons of adjacent cells within tissues. In addition to these well-studied types of AJs, most epithelial cells possess a third, largely neglected

class of junctions located in the mid-lateral region of cell–cell contacts, generally referred to as lateral AJs (also termed spot-like AJs or puncta adhaerentia). These submicron-sized junctions also associate with actin filaments but, unlike apical and basal AJs, they neither recruit vinculin or afadin, two key actin-binding proteins of apical and basal AJs, nor they show any directional movement (Indra et al., 2013; Kametani and Takeichi, 2007; Wu et al., 2014a; Wu et al., 2014b). Instead, lateral AJs are specifically enriched with signaling proteins such as erbin (Choi et al., 2019) and PLEKHA5 (Sluysmans et al., 2021). Collectively, these observations, along with functional experiments reported by Wu et al. (2014b), suggest that these AJs may play a role in intercellular signaling. The molecular machinery underlying this AJ diversification remains largely unexplored.

A key structural unit common to all types of AJs is the cadherin–catenin complex (CCC). The transmembrane component of this complex (E-cadherin in epithelia) belongs to the family of classical cadherins. The intracellular tail of these cadherins simultaneously binds two Armadillo (ARM)-repeat proteins (Franke, 2009; Mège and Ishiyama, 2017; Takeichi, 2014). The distal C-terminal region interacts with the 12 ARM repeat protein, β-catenin (or its ortholog, plakoglobin), which links cadherin to the actin-binding protein, α-catenin. The juxtamembrane region of the cadherin tail interacts with 9 ARM repeat protein, p120-catenin (p120) or with its orthologs, plakophilin 4 (pkp4), δ2-catenin, or ARM repeat protein deleted in velo-cardio-facial syndrome (ARVCF). These four proteins comprise a distinct

[1]Department of Dermatology, The Feinberg School of Medicine, Northwestern University, Chicago, IL, USA;   [2]Center for Advanced Microscopy/Nikon Imaging Center, Northwestern University, Chicago, IL, USA;   [3]Department of Cell & Developmental Biology, The Feinberg School of Medicine, Northwestern University, Chicago, IL, USA.

Correspondence to Sergey M. Troyanovsky: s-troyanovsky@northwestern.edu.

**Rockefeller University Press**
J. Cell Biol. 2026 Vol. 225 No. 4    e202507134



**https://doi.org/10.1083/jcb.202507134**    1 of 16

δ-catenin subfamily within the 9 ARM repeat plakophilin/δ-catenin protein family. Our previous experiments suggested that the cadherin-associated proteins play a leading role in directing cadherin into distinct types of CCC clusters (Troyanovsky et al., 2021) that could potentially result in formation of functionally and morphologically distinct types of AJs.

Here, we investigated the roles of p120 and pkp4 in AJ diversification. These two proteins evolved from a common ancestor at the onset of vertebrate evolution (Gul et al., 2017; Zhao et al., 2011). Most mammalian epithelial cells co-express p120 and pkp4 along with their relative, ARVCF. Because these proteins bind to the same site of E-cadherin (Mariner et al., 2000), the cells form three distinct CCC variants: p120-CCC, pkp4-CCC, and ARVCF-CCC, with p120-CCC being the predominant form in most epithelia. δ-Catenin proteins are thought to act redundantly in two key functions. They regulate the CCC stability by controlling the accessibility of cadherin endocytic motifs (Cadwell et al., 2016; Davis et al., 2003), and they regulate the strength and dynamics of AJs by controlling the AJ-associated cytoskeleton through Rho GTPase signaling (Anastasiadis and Reynolds, 2001; Keil et al., 2013). It has been suggested that distinct δ-catenins may differentially fine-tune these regulations (Donta et al., 2023; Müller and Hatzfeld, 2025). However, an earlier immunomorphological study from Dr. Franke's laboratory showing that pkp4 is enriched only in a subset of AJs (Hofmann et al., 2008) suggested that δ-catenins may contribute to AJ diversification. Our immunolocalization and functional analyses of p120 and pkp4 provide direct evidence supporting this hypothesis. Our data show that these proteins play roles in the formation of distinct AJ subtypes by facilitating specialized assembly pathways. While p120-CCC supports the formation of apical and basal AJs, where CCC clustering relies on α-catenin–actin interactions, pkp4-CCC promotes the formation of lateral AJs, where CCC clustering is reinforced by a novel, α-catenin–independent mechanism. Thus, the interplay between canonical, p120-directed, and noncanonical, pkp4-directed, pathways of AJ assembly contributes to the spatial organization of epithelial cell–cell adhesions.

## Results

### Pkp4 is predominantly recruited into lateral AJs

Epidermal A431 cells, like most human epithelial cells, simultaneously express three δ-catenins and therefore assemble three distinct CCC species: p120-CCC, pkp4-CCC, and ARVCF-CCC. Our proteomics analysis of A431 cells suggests that the latter two complexes together account for only ~20% of all CCCs (Troyanovsky et al., 2021). In this study, we focused on p120-CCC and pkp4-CCC. Triple immunostaining of A431 cells for pkp4, p120, and E-cadherin showed that pkp4-CCC did not fully colocalize with p120-CCC. Instead, pkp4 was enriched in distinct AJs or appeared in separate clusters within larger AJs (Fig. 1, a and b). Detailed inspection of basal, middle, and apical confocal sections, together with evaluation of the pkp4 fluorescence intensity relative to E-cadherin across AJs of different locations (pkp4/E-cad index), revealed a striking pattern: pkp4 was most prominent in AJs located in the mid-lateral regions of cell–cell contacts (Fig. 1 c and Fig. S1 a). Based on this mid-lateral position and characteristic

dot-like appearance, these AJs correspond to AJs known as lateral AJs. In contrast, p120, although also present in lateral AJs, was especially enriched in apical and basal AJs. This spatial segregation of the two CCC species was further highlighted by scatterplot analysis of pixel intensities (Fig. 1 b, right). The linear relationship between p120 and E-cadherin was high (Pearson's r ~ 0.85), whereas correlations between pkp4 and E-cadherin (r ~ 0.55) or between pkp4 and p120 (r ~ 0.45) were substantially lower. In the latter case, only a small fraction of pkp4- and p120-derived pixels showed interdependence. These observations were corroborated by colocalization color maps of E-cadherin–p120-pkp4–stained cells, which enabled both quantitative and visual assessment of colocalization (Fig. S1 b). Finally, consistent with published data (Hofmann et al., 2008), no colocalization was detected between the desmosomal cadherin, desmoglein-2 (dsg2), and pkp4 (Fig. S1, c and d).

To determine whether preferential recruitment of pkp4-CCC to lateral AJs is a general feature of epithelial cells, we examined DLD1 colon carcinoma cells. Double staining for pkp4 and p120 showed that pkp4 was also enriched in lateral AJs in these cells, whereas p120 was the dominant species in apical AJs (Fig. 1 d), which in DLD1 cells belong to the linear apical AJ subtype associated with the circular actin bundle (Choi et al., 2019). Accordingly, scatterplots of pkp4 and p120 pixel intensities, collected from apical versus lateral focal planes (Fig. 1 e), displayed two distinct linear relationships (r ~ 0.55 and ~0.75, respectively).

### Lateral AJs and apical/basal AJs incorporate distinct sets of proteins

Previous characterization of apical AJs in A431 cells has shown that these AJs are enriched in tension-dependent actin-binding proteins such as vinculin and afadin (Indra et al., 2013; Indra et al., 2020). Thus, to validate that p120 and pkp4 preferentially associate with different AJ types, we examined their colocalization with these tension-sensitive proteins. Visual inspection of the stained cells confirmed that vinculin and afadin were predominantly associated with p120-enriched AJs (Fig. 2 a). Consistently, scatterplots of red (vinculin or afadin) versus green (p120 or pkp4) pixel intensities showed that a substantial fraction of pixels derived from the tension-dependent proteins correlated with p120-derived pixels, whereas correlations with pkp4 were markedly lower (Fig. 2 c). Accordingly, these two proteins showed a relatively low Pearson's correlation value with pkp4 (r < 0.5), while it was above 0.5 in the case of p120 (Fig. 2, c and e).

Although the specific protein composition of the lateral AJs had not been accurately characterized, available data suggest that they are enriched with erbin (Choi et al., 2019) and PLEKHA5 (Sluysmans et al., 2021). We therefore tested the colocalization of pkp4 and p120 with these two proteins. Double staining showed clear colocalization of PLEKHA5 with pkp4 in lateral AJs (Fig. 2 b), which produced a strong pixel-intensity correlation between these proteins (r ~ 0.76; Fig. 2 d). While erbin displayed a more uniform distribution, both visual inspection and scatterplots also indicated some preference for pkp4-enriched AJs (Fig. 2, d and e). Altogether, these results suggest that p120-CCC and pkp4-CCC preferentially associate with

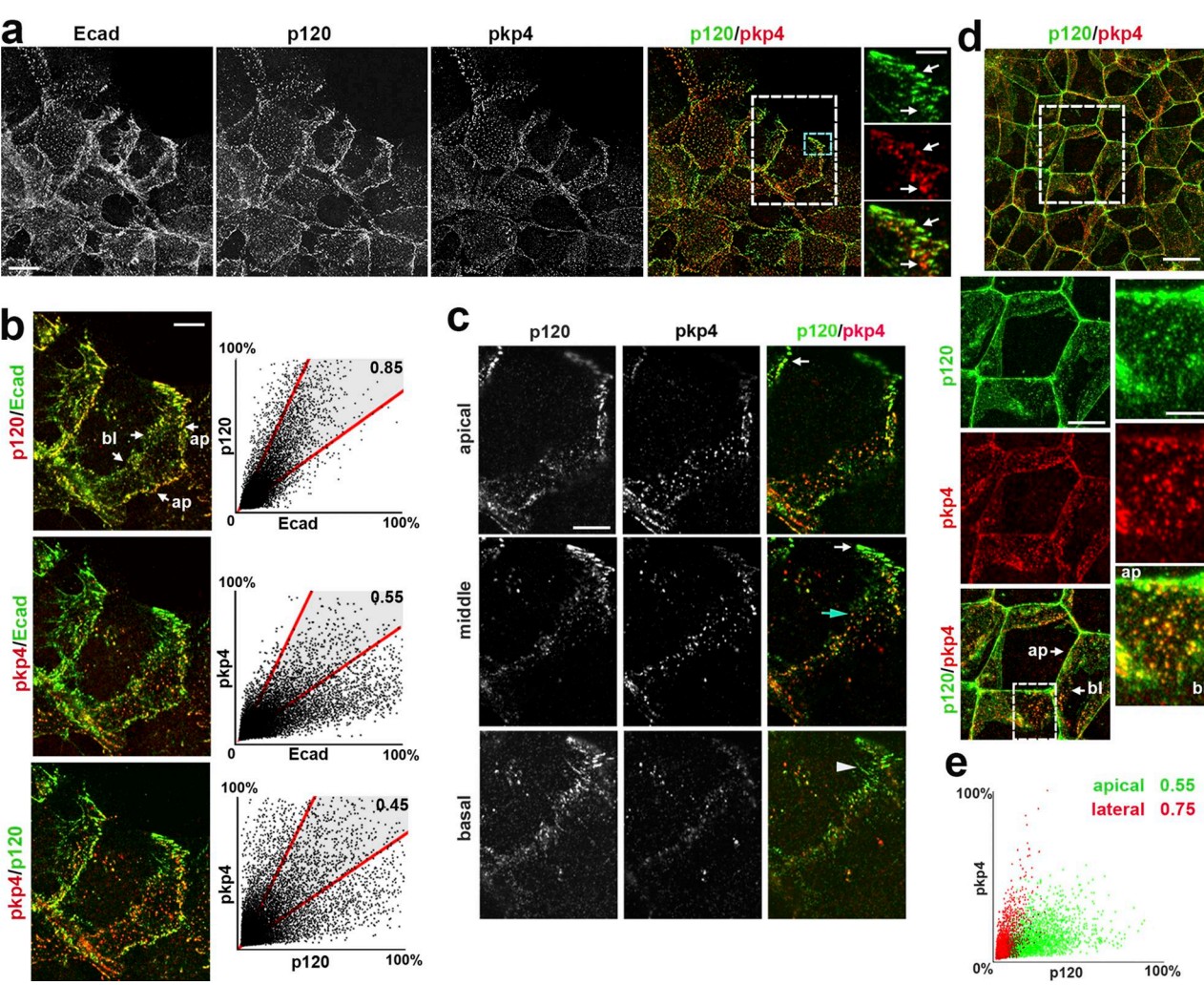

Figure 1. **Lateral and apical AJs in A431 and DLD1 cells recruit different combinations of p120 and pkp4. (a)** Projections of all x–y optical slices of A431 cells triple stained for E-cadherin (Ecad), p120, and pkp4. The merged image shows p120 (green) and pkp4 (red). Bar, 20 µm. The region outlined by the blue dashed box is enlarged at the right. Arrows highlight pkp4-enriched clusters, which either completely lack p120 (bottom arrow) or be embedded within p120-rich apical AJs (top arrow). Bar, 5 µm. **(b)** The region marked by the white dashed box in a is shown in three staining combinations: p120 (red)/Ecad (green), pkp4 (red)/Ecad (green), and pkp4 (red)/p120 (green). Note that p120 and E-cadherin are present in nearly all AJs, whereas pkp4 is predominantly enriched in the submicron size dot-like AJs. Scale bar, 20 µm. Apical (*ap*) and basal (*bl*) ends of the lateral membranes are indicated. Corresponding scatterplots of relative pixel intensities are shown on the right; Pearson's r values appear in the upper right of each plot. Note that p120/Ecad pixels cluster within a region of strong positive correlation (shaded zone between red lines), whereas pkp4/Ecad and pkp4/p120 show weaker correlations. **(c)** Projections of optical z-slices spanning the apical, middle, and basal regions of the cell shown in b. The apical projection spans the top 0.9 µm, the middle projection the next ~1.5 µm, and the basal projection—the bottom 0.6 µm of the cell. Bar, 20 µm. Note that p120 is concentrated at apical AJs (white arrows) and basal AJs (arrowhead), whereas pkp4 is enriched in dot-like AJs in the middle portion of the contact (one example indicated by a blue arrow). **(d)** Maximum-intensity projection of all x–y slices of DLD1 cells stained for p120 (green) and pkp4 (red). Only the merged image is shown. Bar, 20 µm. A zoom of the dashed box is shown at bottom left (bar, 10 µm), and a further magnified view of its dashed region is shown at the right (bar, 5 µm). **(e)** Scatterplots of p120/pkp4 relative pixel intensities for apical (green) and lateral (red) optical slices of the cell shown in d. Pearson's r values for each pixel population are indicated in the upper right.

distinct AJ subtypes: p120-CCC with apical and basal AJs and pkp4-CCC with lateral AJs.

### Lateral AJs are persistent but mobile structures

The absence of vinculin or afadin, the proteins that are important for reinforcing CCC-actin bonds, in lateral AJs suggested that these junctions might be unstable. To test this, we performed time-lapse imaging of A431 cells in which endogenous E-cadherin was replaced with mGFP-tagged E-cadherin (EcGFP). 1-hour movies acquired at 30-s intervals showed that lateral AJs were mobile yet surprisingly stable: their majority persisted

throughout the entire 1-h-long observation period (Fig. 3, a and b; and Video 1). Tracking their trajectories (Fig. 3, b and c) and mean square displacement (MSD) analyzes (Fig. S1 f) revealed that lateral AJs displayed oscillatory, active movements consistent with previously reported lateral AJ dynamics in other epithelial cells, likely driven by myosin II–mediated contractile activity of the actin cortex (Wu et al., 2014a). These movements lacked a defined overall direction, although neighboring junctions often exhibited coordinated motion (red tracks in Fig. 3 c). Fusion and fission events were also frequently observed (Fig. 3 a). Lateral AJs were positioned immediately beneath a row of

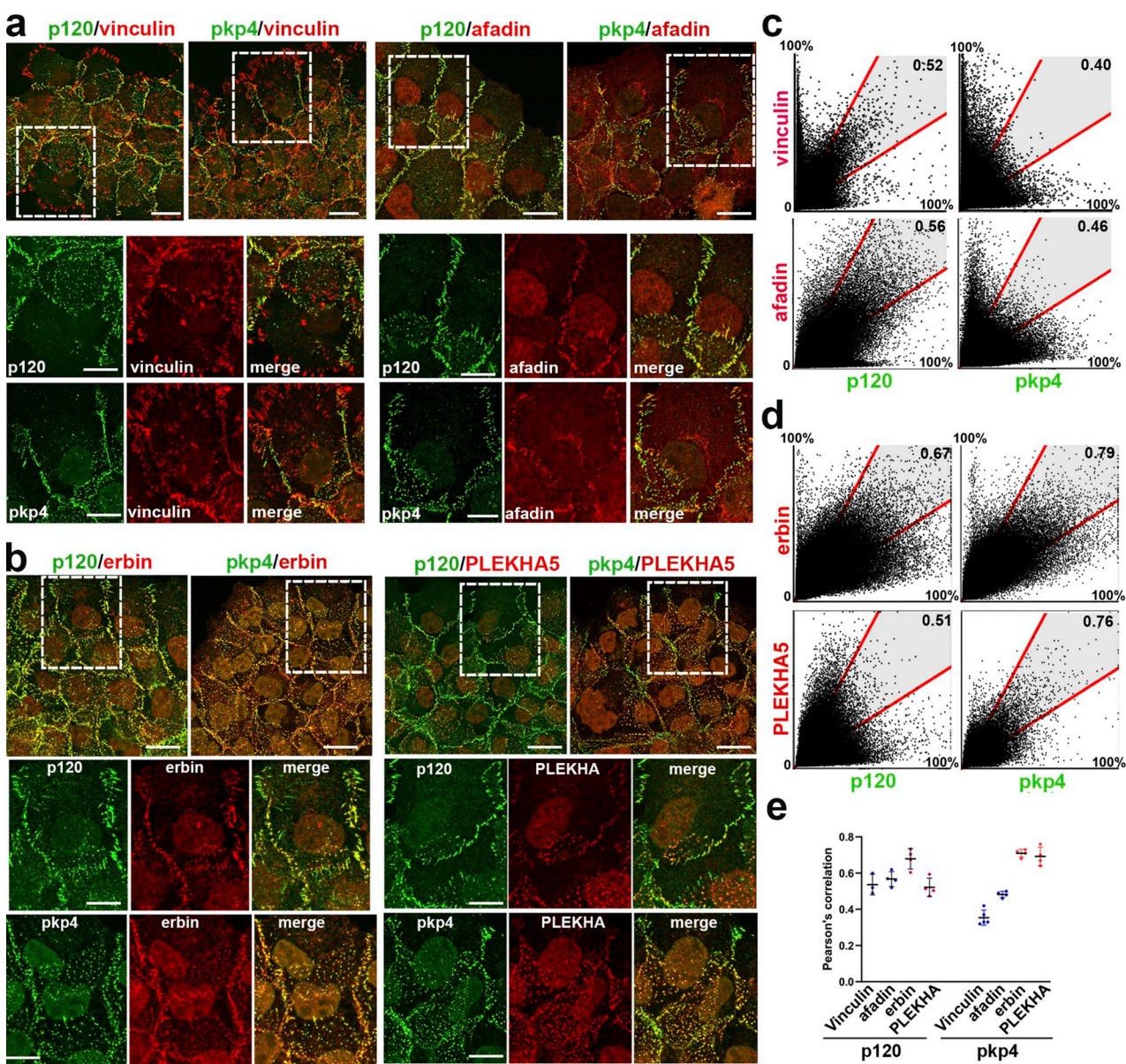

Figure 2. **Pkp4- and p120-enriched AJs associate with different sets of proteins. (a and b)** Maximum-intensity projections of all x–y optical slices of A431 cells double-stained for p120 or pkp4 (green) in combination with apical/basal AJ markers, vinculin, and afadin (a) or lateral AJ markers, erbin, and PLEKHA5 (b). Only merged images are shown for low magnifications. Bar, 20 µm. The zoomed regions indicated by white dashed boxes are shown in both channels at the bottom. Bar, 12 µm. **(c and d)** Corresponding scatterplots of red and green relative pixel intensities for the images in a and b. The Pearson's r values are shown at the upper right of each plot. Note that substantial populations of p120/vinculin and p120/afadin pixels exhibit strong positive correlation indicated by shaded area between the red lines. Such pools of positively correlated pixels are undetectable for pkp4/vinculin or pkp4/afadin combinations. Instead, these combinations display two independent pixel populations distributed along the x and y axes. By contrast, pkp4 shows a stronger relationship with the lateral AJs markers, PLEKHA5 and erbin. **(e)** Average Pearson's colocalization values (r) of images stained as in a and b. Four independent images were quantified for each marker pair. The means ± SD are indicated by bars. Note that apical AJ markers (vinculin and afadin) show substantially higher r values with p120 than with pkp4, whereas the opposite trend is observed for the lateral AJ markers (PLEKHA5, erbin).

relatively immobile apical AJs and above the region occupied by basal AJs. Consistent with published data (Hong et al., 2010; Kametani and Takeichi, 2007), basal AJs were continuously generated at the basal end of the lateral plasma membrane and then moved upward (green tracks in Fig 3 c). Most basal AJs were relatively short-lived because, during their upward movement, they either merged with apical AJs or entered the region occupied by the lateral AJs, where they eventually disappeared (both

examples shown by green tracks in Fig. 3 c). Occasionally, basal AJs that initially moved upward transitioned into a more non-directional motion typical for lateral AJ. Whether these rare events represent interconversion of basal AJs into lateral AJs and if so, what mechanism might underlie such interconversion, remains unclear. Taken together, our imaging data show that the lateral membranes of A431 cells could be arbitrarily divided in three zones, each containing a specialized type of AJs (Fig. 3 c).

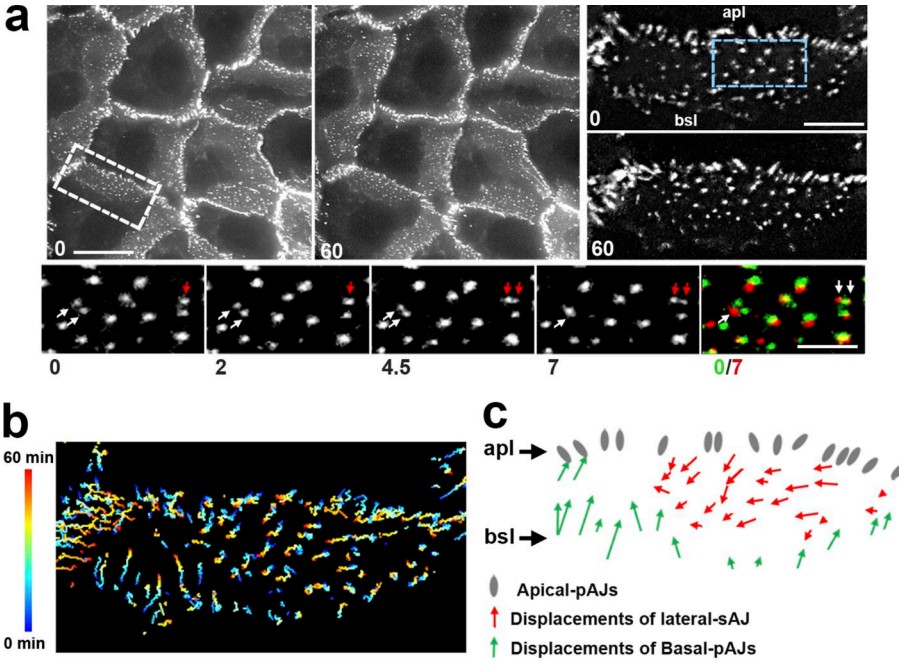

Figure 3. **General topography and dynamics of AJs in A431 cells. (a)** Time-lapse images of EcGFP-expressing A431 cells acquired at 30-s intervals. Only the first (0 min) and final (60 min) frames of the 1-h sequence are shown. Bar, 20 µm. The zoomed cell–cell contact (indicated by dashed white box) from the same two frames is shown on the right (see Video 1). The apical (apl) and basal (bsl) edges of the contact are indicated. Bar, 7 µm. The bottom row shows a 7-min subset of Video 1 illustrating a fusion (white arrows) and a fission (red arrows) events occurring within the region marked by the blue dashed box (times in minutes are indicated). The rightmost image overlays the initial (green) and final (red) frames of this subsequence. Bar, 4 µm. **(b)** Tracked trajectories of major AJs within the region indicated by the white dashed box in a. The color code of trajectories corresponding to time is shown on the left. **(c)** Schematic representation of the net displacements of major AJs derived from the trajectories shown in b. Note that the lateral membrane is clearly partitioned into three zones containing apical AJs (gray), lateral AJs (red), and basal AJs (green). The lateral AJs exhibit nondirectional motion, while basal AJs move upward; two of them ultimately merge with apical AJs. The positions of apical AJs are shown approximately.

## Pkp4 and p120 support distinct types of AJs

To test the roles of pkp4 and p120 in AJ diversification, we knocked out pkp4 or p120 from our EcGFP-expressing cells. As expected, p120 knockout resulted in a dramatical reduction (up to 90%) of EcGFP expression (Fig. 4 a), whereas EcGFP levels were unchanged in pkp4-KO cells. Despite these differences, both p120-KO and pkp4-KO cells still formed AJs (Fig. S2 a). Moreover, the strong linear relationship between EcGFP and pkp4 in p120-KO cells and between EcGFP and p120 in pkp4-KO cells (r ∼ 0.9) indicate that nearly all AJs in these cells were built from pkp4-CCC or p120-CCC, respectively (Fig. S2 b).

To determine the types of AJs formed in the absence of p120 or pkp4, the cells were stained for vinculin and PLEKHA5, the best available markers for apical/basal AJs and lateral AJs, respectively (Fig. 4 b and Fig. S2 c). It showed that both knockout lines produced both types of AJs. However, their abundance was knockout specific. In p120-KO cells, many apical AJs and nearly all basally located AJs displayed little or no vinculin staining (Fig. S2 c) but were positive for PLEKHA5 (Fig. 4 b), suggesting a strong enrichment of lateral AJs. Conversely, PLEKHA5 was only rarely detected in pkp4-KO AJs (Fig. 4 b), whereas vinculin was detectable even at some junctions positioned in the middle region of the lateral membrane (Fig. S2 c). This imbalance between lateral and apical/basal AJs in the KO cells was confirmed quantitatively. We estimated the relative abundance of each AJ type by calculating the ratio of heterochromatic pixels positive for E-cadherin/PLEKHA5 (lateral AJs) or E-cadherin/vinculin (apical/basal AJs) to the total number of E-cadherin–positive pixels (all AJs). Lateral AJ abundance decreased by ∼70% in pkp4-KO cells but increased by ∼50% in p120-KO cells relative to parental EcGFP-expressing cells. In contrast, apical/basal AJ abundance rose in pkp4-KO cells and decreased in p120-KO cells (Fig. 4 c).

Time-lapse imaging revealed even more dramatic differences between p120-KO and pkp4-KO cells. In pkp4-KO cells, most AJs at the lateral membrane exhibited the characteristic upward movement of basal AJs (Fig. 4, d and f; and Video 2). These cells also showed marked instability of AJs at the lateral membrane: lateral AJs frequently and rapidly disappeared. This process typically began with conversion of the junction into a small fluorescent "cloud." Fig. 4 d (bottom row) highlights four such disappearance events occurring within just the first 2 min of a 60-min recording (frames 0–4 of Video 2) in a small region of the lateral membrane. In contrast, p120-KO cells displayed no sustained upward AJ movement (Fig. 4, e and f; and Video 3). Like parental EcGFP cells, their lateral AJs were stable and persisted throughout the entire imaging periods (bottom row, Fig. 4 e). Taken together, these data suggested that pkp4-CCC and p120-CCC, while able to support all types of AJs, are optimized for lateral AJs and apical/basal AJs, correspondingly.

## Pkp4 and p120 maintain opposite AJ dynamics

One explanation for these results is that p120, but not pkp4, supports rapid CCC turnover, thereby destabilizing lateral AJs while promoting the directional movement of basal AJs. To test this idea, we performed FRAP assay. The individual AJs located at the lateral membrane were photobleached, and their fluorescence recovery was monitored over 6 min (Fig. 4 g). Consistent with our hypothesis, p120-deficient AJs were exceptionally stable: they recovered only ∼20% of their fluorescence during the observation period. In contrast, pkp4-deficient junctions were the most dynamic: they reached nearly complete recovery

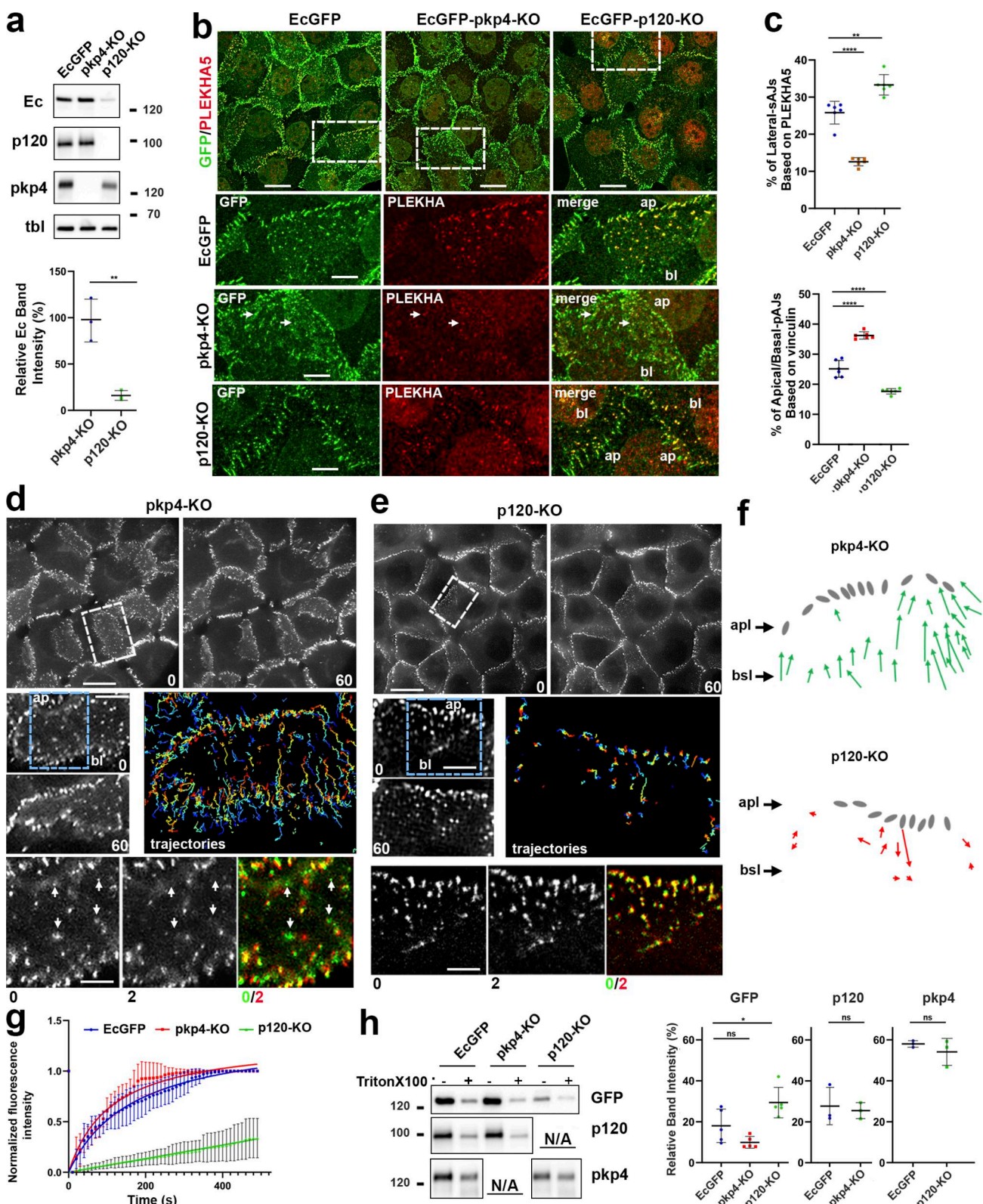

Figure 4. **Changes in AJs upon p120 and pkp4 knockout. (a)** Western blot of A431-EcGFP cells (EcGFP) and their pkp4-KO and p120-KO derivatives probed for EcGFP (Ec), p120, and pkp4. β-Tubulin staining (tbl) serves as a loading control. Molecular weight markers (in kDa) are on the right. The EcGFP band intensities in the KO cells (relatively to the parental EcGFP cells) are quantified at the bottom. **(b)** Projections of all x–y optical slices of the same cells stained for EcGFP (GFP, green) and PLEKHA5 (PLEKHA, red). Low-magnification panels show merged images only. Bar, 20 μm. Zoomed regions (dashed boxes) are shown below in both channels. (*ap* and *bl* indicate apical and basal ends of the lateral membranes). Bar, 12 μm. Arrows indicate a few remaining PLEKHA5-positive AJs in pkp4-KO cells. **(c)** Quantification of heterochromatic EcGFP/PLEKHA5-positive pixels (lateral AJs) or EcGFP/vinculin-positive pixels (apical/basal AJs), expressed as fractions of total EcGFP-positive pixels (*n* = 6 images from three experiments). Means ± SD are shown. **(d and e)** Time-lapse imaging of EcGFP-pkp4-

KO (d) and EcGFP-p120-KO (e) cells at 30-s intervals. Only the first (0 min) and last (60 min) frames are shown (see Fig. 3). Bar, 20 μm. Corresponding zoomed regions (white dashed boxes) are shown below (see Videos 2 and 3). Apical (ap) and basal (bl) edges are indicated. Additional zoomed areas of these contacts (blue dashed boxes) taken 2 min apart (bottom row) illustrate lateral AJ instability in pkp4-KO cells and stability in p120-KO cells. The overlay (right) shows the first frame in green and the second in red; arrows mark AJs that disassembled. Bar, 4 μm. Tracked AJ trajectories are shown as in Fig. 3 b. **(f)** Schematic representation of net displacements of major AJs according to the trajectories in d and e. The AJs are depicted as in Fig. 3 c. **(g)** FRAP analysis of lateral AJs in cell lines indicated as in a (n = 15; mean ± SD). **(h)** Left: Western blot of total cell lysates from control cultures (–) and parallel cultures extracted for 5 min with 1% Triton X-100 (+), probed for EcGFP, p120, and pkp4 (cells are indicated as in a) Right: Quantification of EcGFP intensities in the extracted cells relative to the corresponding non-extracted controls (five independent experiments). Statistical significance for all graphs was calculated using two-tailed Student's *t* tests: ns, nonsignificant; *P < 0.05; **P < 0.01; ****P < 0.0001. The means ± SD are indicated by bars. Source data are available for this figure: SourceData F4.

(∼100%) within ∼5 min, with a recovery halftime (t1/2) of ∼1 min. WT lateral AJs exhibited an intermediate phenotype (t1/2∼2 min).

In parallel experiments we assessed the anchorage of p120-CCC and pkp4-CCC to the cytoskeleton by testing their resistance to Triton X-100 extraction. To this end, the levels of E-cadherin, p120, and pkp4 remaining in cells after detergent extraction were compared with those in untreated cells. The results showed that ∼20% of EcGFP and p120 remained bound to the control and pkp4-KO cells after Triton treatment (Fig. 4 h). This Triton X-100–resistant EcGFP fraction was slightly higher (∼30%) in p120-KO cells. Notably, the Triton-resistant fraction of pkp4 was significantly higher (∼ 60%) in both control and p120-KO cells. Assuming that pkp4 levels correspond to the amount of pkp4-CCC, these results indicate that pkp4-CCC is more tightly bound to the cytoskeleton than p120-CCC.

### Recombinant pkp4 and p120 facilitate formation of lateral AJs and apical/basal AJs, respectively

The differences in AJ phenotypes between control, pkp4-KO, and p120-KO cells suggested that pkp4 reinforces lateral AJs. However, the high abundance of stable lateral AJs in p120-KO cells could also be explained by the markedly reduced EcGFP level in these cells (see Fig. 4 a), which may indirectly affect cadherin turnover in AJs. In addition, ARVCF, the third δ-catenin expressed in A431 cells, complicates the straightforward interpretation of the knockout results. To decisively demonstrate that pkp4 and p120 play specific roles in the formation and/or maintenance of distinct types of AJs, we generated A431 cells lacking all three endogenous δ-catenins (p120, pkp4, and ARVCF) and then re-expressed GFP-tagged pkp4 (GFPpkp4) or GFP-tagged p120 (GFPp120) in these δ-catenin null cells (A431-δCat-KO cells). Western blot analysis (Fig. 5 a) showed that A431-δCat-KO cells retained only trace amounts of E-cadherin, but as expected, its level increased upon GFPpkp4 expression, although not to the level of WT cells. Notably, GFPpkp4 was strongly overexpressed in GFPpkp4-δCat-KO cells relative to endogenous pkp4 in WT A431 cells. However, anti-GFP staining of the same western blots showed that this high GFPpkp4 expression level was still considerably lower than the level of GFPp120 in GFPp120-δCat-KO cells (Fig. 5 a).

Consistent with the data obtained from EcGFP-expressing p120-KO cells, GFPpkp4-expressing δCat-KO cells formed numerous prominent lateral AJs (Fig. 5 b and Fig. S3). The almost perfect co-localization between GFPpkp4 and E-cadherin (r ∼ 0.9) confirmed that virtually all AJs in these cells were composed of GFPpkp4-CCC (Fig. S3). As observed in p120-KO cells, lateral AJs and even some of AJs formed at the basal or apical edges of

the lateral membrane (e.g., at sites of basal or apical AJs) showed no vinculin staining (Fig. 5 b). Accordingly, quantification of the heterochromatic vinculin/GFP and PLEKHA5/GFP pixels revealed that GFPpkp4-expressing δCat-KO cells, like p120-KO cells and unlike the control EcGFP-expressing cells, produced an increased pool of lateral AJs at the expense of apical/basal AJs (Fig. 5, b and d). Time-lapse imaging (Video 5) and FRAP analysis supported this interpretation: AJs of GFPpkp4-expressing cells showed no directional upward movement (Fig. 5, e and g) and exhibited extremely slow fluorescence recovery (Fig. 5 h). Thus, cells expressing GFPpkp4 in the absence of other δ-catenins overproduced lateral AJs.

Parallel experiments with GFPp120-expressing δCat-KO cells revealed the opposite phenotype, recapitulating the AJ characteristics of pkp4-KO cells. Specifically, these cells showed a dramatic reduction in PLEKHA5/GFP co-localization together with an increase in vinculin/GFP heterochromatic pixels (Fig. 5, c and d), indicating a strong shift toward apical/basal AJs. Time-lapse imaging further confirmed this conclusion: GFPp120-expressing cells produced predominantly short-lived, upward-moving AJs that frequently disintegrated through a transient "cloud-like" intermediates (Video 4 and Fig. 5, f–h). Consistent with these observations, FRAP analysis revealed strikingly faster turnover rates for GFPp120-containing AJs versus those in GFPpkp4-expressing cells (Fig. 5 h). Collectively, these experiments demonstrate that p120 and pkp4 exert opposite and highly specific effects on the organization of the cell–cell adhesion system and the types of AJs.

### Pkp4-CCC clusters are formed along actin filaments in α-catenin–independent manner

A defining feature of the lateral AJs is the absence of vinculin and afadin, two proteins that associate with the CCC through the tension-dependent conformational changes in α-catenin and whose recruitment is known to reinforce AJs (Choi et al., 2016; Ishiyama et al., 2018; Sakakibara et al., 2020; Seddiki et al., 2018). On the other hand, lateral AJs exhibit exceptional stability coupled with actomyosin-dependent oscillatory movements (Wu et al., 2014a; Wu et al., 2014b), properties that would be expected to promote vinculin and afadin recruitment. To resolve this apparent paradox, we hypothesized that pkp4-CCC might engage actin filaments through an α-catenin–independent mechanism. To test this idea, we used α-catenin–deficient A431 cells (αCat-KO) expressing an mCherry-tagged α-catenin mutant, αCatCH-Δ259, which lacks the C-terminal region of the protein (aa 260–906). This extensive deletion removes all known α-catenin determinants required for direct or indirect α-catenin–actin

Figure 5. **GFPpkp4 and GFPp120 promote formation of distinct types of AJs. (a)** Western blot of WT A431, δCat-KO derivatives, and their GFPpkp4- or GFPp120-expressing variants probed for E-cadherin, p120, pkp4, ARVCF, and β-tubulin. Molecular weight markers are as in Fig. 4 a. Both GFPp120 and GFPpkp4 increase E-cadherin levels. Also note that GFPpkp4 is strongly overexpressed yet remains below GFPp120, whose expression approximately matches endogenous p120 in WT A431 cells. **(b and c)** Maximum-intensity x–y projections of δCat-KO cells expressing GFPpkp4 (b) and GFPp120 (c) stained for GFP (green) and vinculin or PLEKHA5 (red). Low-magnification panels show merged images only. Bar, 20 µm. Enlarged regions (marked by dashed boxes) are shown below in individual channels. Bar, 15 µm. Arrows indicate several sparse PLEKHA5-positive AJs in GFPp120-expressing cells. **(d)** Quantification of GFP/PLEKHA5 or GFP/vinculin heterochromatic pixels (representing sums of lateral and apical/basal AJs, respectively) expressed relative to total GFP signal (n = 6 from three independent images). Note, GFPp120 and GFPpkp4 exert opposite effects on lateral versus apical/lateral AJ abundance. Statistical analysis: two-tailed Student's t test: ****, P <0.0001. The means ± SD are indicated by bars. **(e and f)** Time-lapse imaging of GFPpkp4 (e) and GFPp120 (f) cells acquired at 30-s intervals; 0- and 60-s frames are shown. Corresponding enlarged contact regions (dashed white boxes) are shown at the bottom (see also Videos 4 and 5). Apical (ap) and basal (bl) contact edges are indicated. Bars: 20 µm (top), 10 µm (bottom). **(g)** Representative trajectories of AJs (left) and schematic summaries of their net displacements (right) presented as in Fig. 3. **(h)** FRAP analysis of lateral AJs in the cell lines shown in a (n = 15; mean ± SD). Source data are available for this figure: SourceData F5.

interactions while preserving its N-terminal β-catenin–binding domain (Fig. 6 a).

Western blotting confirmed that αCatCH-Δ259–expressing cells had E-cadherin, p120, and Pkp4 levels comparable with those of WT A431 cells, and that expression of the mutant approximately matched endogenous α-catenin (Fig. 6 b). As expected, both αCatCH-Δ259 and E-cadherin in these cells were broadly distributed along the plasma membrane (Fig. 6 c). However, in addition to this diffuse localization, we observed arrays of small clusters containing both proteins along filopodia-like protrusions connecting adjacent cells. Remarkably, pkp4 was strongly and selectively enriched in these clusters (Fig. 6, c and d). Line-scan analysis showed that, unlike the mutant or E-cadherin, pkp4 was barely detectable outside the clusters. Quantification of the fluorescence inside versus outside the clusters confirmed this observation: while E-cadherin and αCatCH-Δ259 exhibited approximately fourfold enrichment in clusters, pkp4 showed ~10-fold enrichment (Fig. 6 e). In contrast, p120 displayed a much lower cluster/extra-cluster ratio (~3) and was barely detectable in some clusters (Fig. 6, c–e).

We next tested whether the αCatCH-Δ259/pkp4 clusters might arise through vinculin, which can directly interact with β-catenin in some contexts (Hazan et al., 1997; Morales-Camilo et al., 2024; Peng et al., 2010). However, vinculin staining showed no association of this protein with the clusters (Fig. S4 a). Finally, staining for erbin and PLEKHA5 confirmed that the αCatCH-Δ259/pkp4 clusters display the molecular signature of lateral AJs (Fig. S4 b). Altogether, these data strongly suggest that pkp4-CCC forms clusters via a mechanism that is not based on α-catenin–dependent CCC binding to actin.

The formation of αCatCH-Δ259/pkp4 clusters along actin-rich cell–cell protrusions further suggested that despite being α-catenin–independent, these clusters still use actin filaments for their formation. To test this, we gently extracted cells with 1% Triton X-100 in cytoskeleton-preservation buffer to remove most soluble cytoplasmic proteins, and then stained the Triton-resistant cytoskeleton for mCherry and F-actin or pkp4. The αCatCH-Δ259/pkp4 clusters remained Triton-resistant and were most prominent along actin bundles within cell protrusions. Numerous smaller clusters were also distributed along the actin cortex of overlapping lamellipodia between neighboring cells (Fig. 6 f). To more precisely define the relationship between clusters and actin filaments, we performed platinum replica EM (PREM) combined with anti-mCherry immunogold labeling of Triton X-100–extracted cells. Inspecting the cell replicas, we focused on well spread areas of overlapping actin networks of the neighboring cells, where both the gold particles and individual actin filaments could be easily resolved. Strikingly, immunogold particles were organized in numerous clusters of various sizes, all of which were clearly associated with actin filaments, especially often at the sites where individual filaments merged into multifilament sheets (Fig. 6 g). Together, these experiments show that pkp4-CCC form actin-associated clusters through a mechanism that bypasses α-catenin–dependent actin linkage, distinguishing this process from the tension-dependent assembly of apical/basal AJs.

### Pkp4-facilitated, α-catenin–independent CCC clustering has no role in cell–cell adhesion

Next we sought to determine whether α-catenin–independent pkp4-CCC clusters contribute to cell–cell adhesion. To address this question, we performed a dispase adhesion assay comparing control EcGFP cells with their pkp4-KO and p120-KO counterparts. As expected, after dispase-mediated detachment from the substrate, confluent monolayers of control EcGFP-expressing cells contracted due to intrinsic contractile forces within the epithelial sheet. Quantification of sheet areas revealed that this contraction was not significantly altered in pkp4-depleted cells (Fig. 7 a). Overall cell–cell adhesion strength in these cells was only slightly reduced, as evidenced by a small but statistically significant increase in the number of fragments generated upon mechanical stress applied to the lifted sheets. Thus, the reduction in lateral AJs does not substantially compromise general cell–cell adhesion. As suggested by previous studies (Ireton et al., 2002), dramatic decrease in E-cadherin levels in p120-KO cells severely impaired both cell sheet contraction and cell–cell adhesion strength (Fig. 7 a).

To test directly whether α-catenin–independent pkp4-CCC clusters can support cell–cell adhesion, we generated αCat/pkp2/3-KO A431 cells and their derivatives overexpressing GFPpkp4. In addition to the absence of α-catenin, the αCat/pkp2/3-KO A431 cells were also deficient in two desmosomal plakophilins, pkp2 and pkp3, proteins essential for desmosome (DSM) assembly (Fujiwara et al., 2015; Indra et al., 2021; Todorovic et al., 2014). Consistent with this, dsg2 staining confirmed that αCat/pkp2/3-KO A431 cells were unable to form DSMs (Fig. S5). Removal of DSMs eliminated their contribution to cell–cell adhesion, thereby allowing us to detect even minor increases driven specifically by GFPpkp4 overexpression. Despite the complete absence of α-catenin and DSMs, both endogenous pkp4 in αCat/pkp2/3-KO cells (Fig. S5) and recombinant GFPpkp4 in overexpressing variants (Fig. 7 b) efficiently formed clusters, which were predominantly localized along actin-rich intercellular protrusions. We then confirmed that these clusters were trans-associated across cell–cell contacts by mixing GFPpk4-expressing αCat/pkp2/3-KO cells with αCat-KO cells expressing the αCatCH-Δ259 mutant. All clusters formed between these 2 cell populations were heterochromatic (Fig. 7 c), strongly supporting intercadherin trans interactions within these α-catenin–deficient structures. However, these AJs, even in GFPpkp4-expressing cells, were completely unable to generate contractile forces or sustain measurable intercellular adhesion (Fig. 7 d).

## Discussion

Cadherin-mediated adhesions are driven by two CCC oligomerization processes occurring on opposite sides of the plasma membrane. The "outside" cadherin oligomerization mediates the formation of an adhesive paracrystalline lattice, in which extracellular cadherin domains interact through a combination of trans and cis bonds (Honig and Shapiro, 2020; Laur et al., 2002; Leckband et al., 2024; Wu et al., 2010). The "inside" CCC oligomerization is mediated by α-catenin–actin interactions, which generate linear, actin-bound CCC strands (Mei et al., 2020; Troyanovsky et al., 2025; Xu et al., 2020). Acting synergistically, these two relatively independent outside and inside oligomerization processes produce CCC clusters whose adhesion strength can be adjusted by signaling pathways that regulate

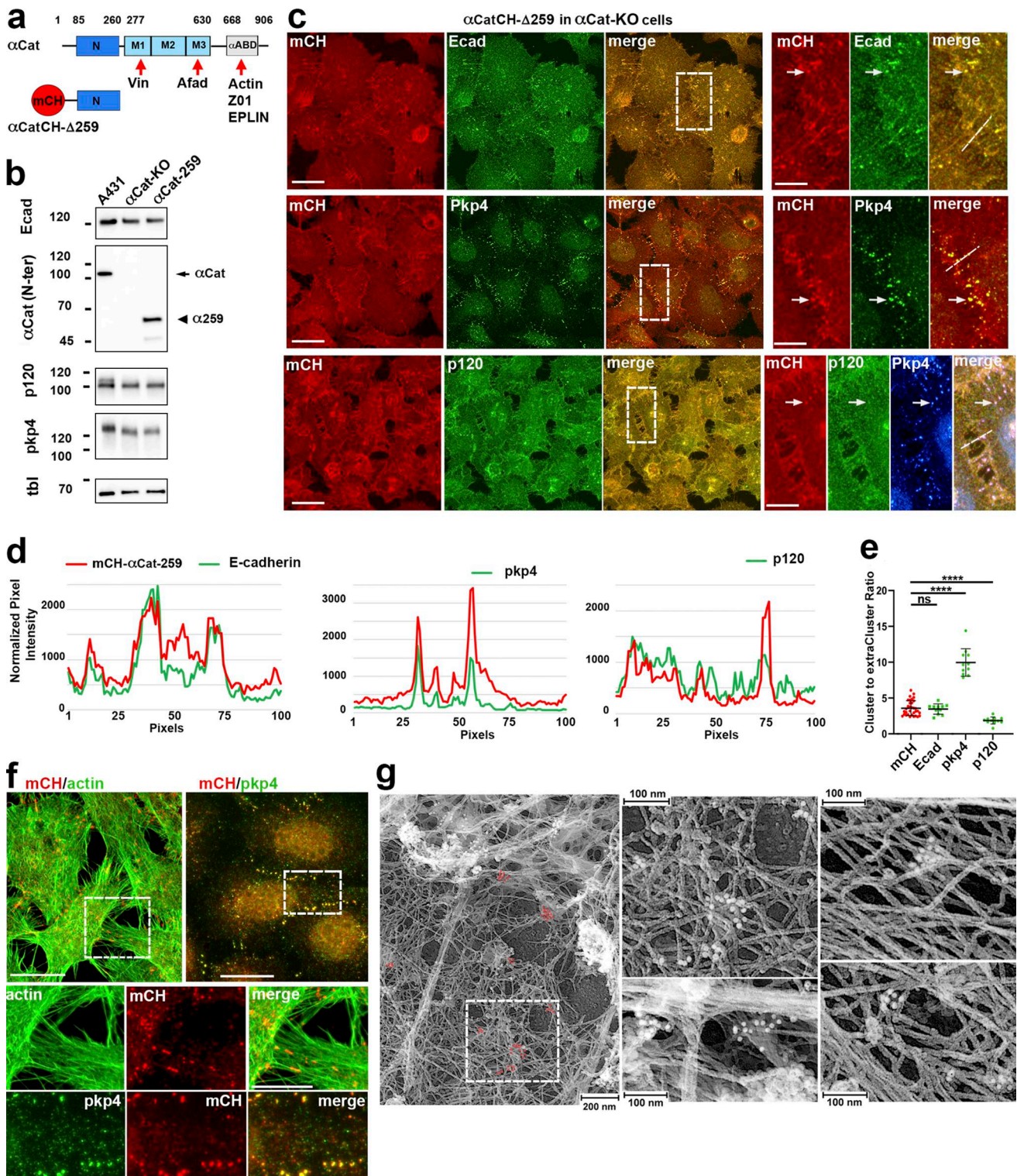

Figure 6. **Pkp4 promotes formation of α-catenin–independent clusters associated with actin filaments. (a)** Schematic representation of full-length α-catenin (αCat) and the truncated mutant αCatCH-Δ259, which contains only the β-catenin–binding N-terminal region (residues 1–259). The extended deletion removes all regions required for direct binding to F-actin (αABD) and for interactions with actin-associated proteins, including vinculin (Vin), afadin (Afad), ZO-1, and EPLIN, located within the middle subdomains (M1–M3). The mCherry tag (mCH) is indicated by a red circle; unstructured regions are shown as solid lines. Domain boundaries are labeled by residue numbers. **(b)** Western blot of WT A431 (A431), αCat-KO cells (αCat-KO), and αCat-KO cells expressing αCatCH-Δ259 (αCat-259) probed for E-cadherin (Ecad), α-catenin N terminus (αCat, N-term), p120, pkp4, and β-tubulin (tbl, loading control). Molecular weight markers (kDa) are shown at left. The positions of intact α-catenin (αCat) and the mutant (α259) are indicated. **(c)** Maximum-intensity x–y projections of αCatCH-Δ259 cells stained for mCH (red) together with E-cadherin (Ecad) or pkp4 (Pkp4), or together with p120 and pkp4 (pkp4 shown only in the zoomed region, blue). Bar, 20 μm. Enlarged regions (dashed boxes) are shown on right. Bar, 6 μm. Arrows indicate representative clusters. **(d)** Line-scan intensity profiles taken along the

dashed line in c. **(e)** Ratios of fluorescence intensities in cluster versus inter-cluster regions for mCH, E-cadherin, pkp4, and p120 (*n* = 11 from two independent images). Statistical analysis: two-tailed Student's *t* test: ns, nonsignificant; ****, P <0.0001. Bars indicate mean ± SD. **(f)** Maximum-intensity projections of αCatCH-Δ259 cells extracted with 1% Triton X-100 and stained for mCH (red) and F-actin or pkp4 (green). Bar, 20 µm. Enlarged regions (dashed boxes) are shown at the bottom rows. Bar, 10 µm. **(g)** PREM of αCatCH-Δ259 clusters. The left panel shows an overview of two overlapping cells; boxed region is enlarged at top middle. Additional panels show representative clusters from other cells in the same culture. Immunogold particles marking mCH (10 nm) in the overview image are pseudocolored red. Source data are available for this figure: SourceData F6.

α-catenin–F-actin association. Understanding this regulation remains an important direction for future work. However, our current view of CCC clustering in AJs is incomplete. Evidence suggests that the canonical outside and inside oligomerization processes represent only a subset of multiple possible binding cascades that may ultimately generate the remarkable diversity of AJ architectures observed in cells. For example, the structures of invertebrate and vertebrate cadherins show the distinct modes of outside oligomerization (Jin et al., 2012). A novel type of cadherin ectodomain dimerization has been recently identified by cryo-EM (Maker et al., 2022). Depending on the involvement of additional adaptor proteins, α-catenin–actin interactions can generate supramolecular assemblies of varying strength and organization (Gong et al., 2025; Vu et al., 2021). Moreover, junction-associated cadherin clusters have been observed even in α-catenin–deficient cells (Chen et al., 2015; de Groot et al., 2018; van Hengel et al., 1997). Here, we demonstrate that at least a subset of these α-catenin–independent clusters is formed through a pkp4-dependent inside oligomerization mechanism, which contributes to the diversification of AJs.

Using two epithelial models, A431 epidermal carcinoma and DLD1 colon carcinoma, we demonstrate that pkp4, unlike the more abundant δ-catenin family member, p120, is preferentially incorporated into lateral AJs. These AJs reside in the middle section of the lateral membrane and are distinguished from the apical and basal AJs by three defining features: (1) absence of vinculin and afadin (Indra et al., 2013); (2) enrichment of PLE-KHA5 and erbin, although these proteins, particularly erbin, can occasionally appear as isolated clusters in other AJ types (Choi et al., 2019; Sluysmans et al., 2021); and (3) nondirectional, actomyosin-driven oscillatory motion (Wu et al., 2014a; Wu et al., 2014b). Despite their constant movement, our tracking these junctions in live A431 cells shows that they are remarkably long-lived.

Our analysis of pkp4-KO cells shows that although pkp4 is preferentially recruited into lateral AJs, it is not absolutely required for their formation. Nevertheless, pkp4 loss markedly reduces the number of lateral AJs. Furthermore, majority of the remaining AJs at the lateral membrane of pkp4-KO cells exhibit traits of basal AJs. The phenotype appears specific to lateral AJs, as these pkp4-KO cells show no detectable defects in apical or basal AJs, and they maintain near-normal cell–cell adhesion in the dispase assay. The importance of pkp4 for lateral AJ maintenance is further underscored by the fact that p120 depletion that elevates the relative proportion of pkp4-CCC results in the inverse outcome: while some of apical, vinculin-containing AJs remained, majority of AJs in p120-KO cells acquire lateral AJ characteristics. This "instructive" role of pkp4 in AJ specialization is further supported by experiments with δ-catenin-null

cells: GFPpkp4 overexpression in these cells leads to a pronounced and selective expansion of lateral AJs, whereas GFPp120 overexpression in the same cells promotes instead overproduction of apical and basal AJs at the expense of lateral junctions.

Collectively, our comparative gain- and loss-of-function analyses of p120 and pkp4 demonstrate that these proteins promote formation of distinct AJ types. While p120 supports the tension-dependent apical and basal AJs whose inside clustering relies on canonical α-catenin–actin interactions, pkp4 reinforces lateral AJs. The absence of vinculin and afadin at these junctions suggests that their remarkable stability does not rely on vinculin/afadin-mediated reinforcement of α-catenin–actin bonds. Furthermore, our analyses show that the canonical α-catenin–actin interactions are dispensable for lateral AJ assembly. Instead, to reinforce the outside cadherin adhesive lattice these junctions rely on an alternative mechanism of inside cadherin clustering.

Consistent with previous studies (Troyanovsky et al., 2025), A431 cells lacking functional α-catenin show no measurable cell–cell adhesion in the dispase assay and do not form AJs anchored to radial actin bundles. Nevertheless, using confocal imaging and immunogold PREM, we show that these cells assemble pkp4-CCC clusters that appear to engage in trans interactions and associate with the cortical actin cytoskeleton. At the light microscopy level, these clusters recruit PLEKHA5 and erbin confirming their lateral AJ identity. Another distinguished feature of these α-catenin–independent clusters is their minimal incorporation of p120. The striking differences in subcellular localization of p120-CCC and pkp4-CCC in these cells strongly suggest their distinct clustering mechanisms and distinct modes of F-actin engagement. This conclusion is further supported by the differential Triton X-100 solubility of p120 and pkp4 in control, α-catenin-expressing cells. Altogether, our findings show that lateral AJs represent specialized structures relying on a unique inside CCC oligomerization mechanism that may involve an α-catenin–independent mode of F-actin interaction. Whether pkp4 directly drives this α-catenin–independent clustering or acts indirectly as an instructive factor activating cadherin clustering through other CCC-associated proteins remains to be determined. The first possibility is supported by our recent observation that the ARM domain of pkp3, that is the most conserved region of the δ-catenin/plakophilin protein family, directly interacts with actin filaments (Gupta et al., 2023). Thus, our working model is that pkp4 promotes a specific actin-binding mode that drives pkp4-dependent CCC oligomerization in lateral AJs.

The relationship between lateral AJs and DSMs is another important aspect for discussion. These two evolutionary related junctions share several properties, including spot-like morphology, lateral membrane localization, and similar protein

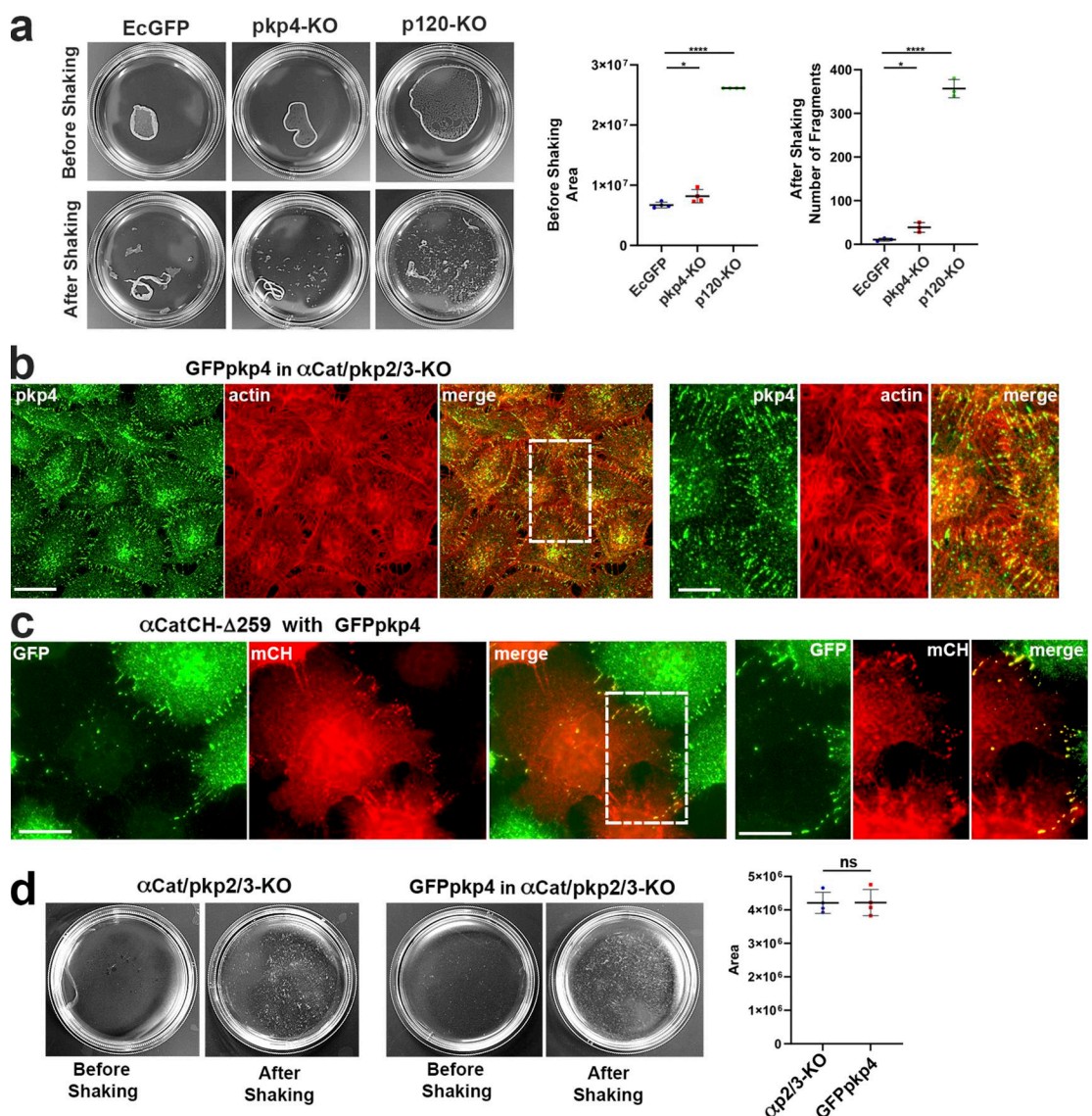

Figure 7. **Pkp4-promoted E-cadherin clusters are ineffective in adhesion. (a)** Dispase-based cohesion assay of A431 cells expressing EcGFP (EcGFP) and their derivative lines lacking pkp4 (pkp4-KO) or p120 (p120-KO). Left: Representative images showing cell sheets detached from the culture dishes before and after application of mechanical stress (shaking). Right: Quantification ($n$ = 4) of sheet areas (pixels) before shaking (left) and the number of sheet fragments generated after shaking (right). Data represent mean ± SD. **(b)** Maximum-intensity projections of all x–y optical slices of αCat/pkp2/3-KO cells expressing GFP-pkp4, stained for pkp4 (green) and F-actin (red). Bar, 20 μm. The boxed region is enlarged at right. Bar, 8 μm. Note the precise alignment of pkp4 clusters along actin-rich cell–cell protrusions. **(c)** Co-culture of αCat/pkp2/3-KO cells expressing GFPpkp4 with αCat-KO cells expressing αCatCH-Δ259. Cells were stained for GFP (green) and mCH (red). Bar, 20 μm. The boxed region is enlarged at right. Heterochromatic GFP/mCH clusters are evident at the interfaces between the two cell populations. **(d)** Dispase-based cohesion assay of parental αCat/pkp2/3-KO cells and the corresponding line expressing GFP-pkp4 (GFPpkp4-αCat/pkp2/3-KO). Left: Representative images showing detached cell sheets before and after mechanical stress. Note that cell sheets from both lines fail to contract. Right: Quantification ($n$ = 4) of sheet areas (pixels) prior to shaking. The number of fragments after shaking was not quantified. Data represent mean ± SD. Statistical significance in a and d was assessed using a two-tailed Student's $t$ test: ns, not significant; *P < 0.05; ****P < 0.0001. αCat, α-catenin; mCH, mCherry tag.

compositions. The N-terminal region of pkp4 interacts with DSM components (Hatzfeld et al., 2003; Setzer et al., 2004), and like lateral AJs, DSMs exhibit exceptional stability and oscillatory motion in living cells (Gloushankova et al., 2003; Windoffer and Leube, 1999). Our finding that pkp4 promotes lateral AJ formation adds another parallel: DSM-specific plakophilins (pkp2 and pkp3) have been shown to drive desmosomal cadherin clustering (Fujiwara et al., 2015; Indra et al., 2021; Todorovic et al., 2014). Understanding how pkp4 promotes lateral AJ

assembly may therefore provide insight into a long-standing question of how DSMs are assembled.

In conclusion, we show that two members of δ-catenin protein family promote the formation of distinct types of AJs. The more abundant δ-catenin, p120, supports cadherin clustering through the canonical α-catenin–dependent mechanism, generating apical and basal AJs that provide strong cell–cell adhesion and are coupled to the actomyosin tensile forces. In contrast, another δ-catenin, pkp4, facilitates the assembly of lateral AJs

through an α-catenin–independent mechanism. Dispase assays of α-catenin–KO cells overexpressing GFPpkp4 suggest that these junctions do not measurably contribute to cell–cell adhesion. We propose that lateral AJs instead function primarily in intercellular signaling by maintaining close apposition between neighboring lateral membranes. Collectively, our findings reveal an unrecognized role of δ-catenins in regulating AJ assembly pathways, thereby controlling the balance among AJ types in cells and ultimately the global architecture of the epithelial cell–cell adhesion system.

## Materials and methods

### Plasmids

The plasmids (all in pRcCMV) encoding GFP-tagged pkp4 were constructed using a cDNA kindly provided by Drs Mechthild Hatzfeld (Martin Luther University Halle-Wittenberg, Halle, Germany) and Kathleen J Green (Northwestern University, Chicago, IL, USA). The plasmid-encoding GFPp120 was obtained using previously published plasmid encoding p120-3A isoform of human p120 (Troyanovsky et al., 2011). The mCherry-tagged mutant of human αE-catenin (αCatCH-Δ259) was constructed using PCR-based mutagenesis of the plasmid encoding the intact protein described previously (Troyanovsky et al., 2025). The general map of the mutant is presented in Fig. 6 a. All plasmid inserts were verified by sequencing.

### Cell culture and transfection

The original DLD1, A431 cells, EcGFP-expressing E-cadherin–deficient A431 cells (EcGFP-EcKO-A431 cells), their p120-KO variant, and α-catenin deficient αCatKO and pkp2/3-deficient pkp2/3KO cells have been previously described (Indra et al., 2018; Indra et al., 2021; Troyanovsky et al., 2021; Troyanovsky et al., 2025). The pkp4-KO version of the EcGFP-EcKO-A431 cells was obtained using the Alt-R CRISPR-Cas9 System (IDT), which had been used in our laboratory to obtain Ec-KO or p120-KO cells (Indra et al., 2018; Troyanovsky et al., 2021). The same strategy was used to obtain δCat-KO-A431 cells. In brief, the cells were transfected with an RNA complex consisting of a gene-specific CRISPR RNA (crRNA; designed by software of the Broad Institute of Harvard and the Massachusetts Institute of Technology) and transactivating RNA. The following crRNAs were used: pkp4-5′-AGGATCAACTAACAACCATG-3′ and ARVCF-5′- CATCCGAAG ATGGCACAACC-3′. The GFPpkp4-, GFPp120-, and αCatCH-Δ259–expressing cells were obtained using stable transfection of the corresponding cells with the corresponding plasmids. The cells were grown in DMEM supplemented with 10% FBS and were transfected using Lipofectamine 2000 (Invitrogen) according to the company protocol. After selection of the Geneticin-resistant cells (0.5 mg/ml), the cells were sorted for transgene expression by FACS, and only moderate-expressing cells were used. At least three clones were selected for each construct, and all were tested in most of the assays. The expression levels and sizes of the recombinant proteins in the obtained clones were analyzed by western blotting as previously described (Troyanovsky et al., 2025) using anti-GFP or anti-mCherry antibodies and Nitrocellulose Blotting Membrane (#10600007; Amersham). All clones of

cells expressing a particular transgene exhibited the same phenotype. Representative data for one of three clones are presented.

The dispase assay was performed as described by Calautti et al. (1998). In brief, confluent cultures of cells grown on 5-cm dishes were incubated with 2.4 U/ml Dispase II (D4693; Sigma-Aldrich) in DMEM at 37°C, for 30 min. Cells lifted from the substrate as an intact cell sheet were imaged and then the sheets were submitted to mechanical stress on a shaker at 60 rpm. For measuring the sheet area and counting the sheet fragments, the ImageJ tools ("cell count" and "Polygon selection") were used.

### Immunofluorescence microscopy

For immunofluorescence, cells were grown for 2 days on glass coverslips or imaging glass-bottom dishes (P35G-1.5; Mattek) and were fixed with 3% formaldehyde (5 min) and then permeabilized with 1% Triton X-100 (15 min), as described previously (Indra et al., 2020). The confocal images were taken using a Nikon AXR laser scanning microscope equipped with a Plan Apo 60×/1.45 objective lens. Immediately before imaging, the dishes were filled with 90% glycerol. The images were then processed using Nikon's NIS-Elements software. For immunostaining and western blotting, the following antibodies were used: Mouse anti-E-cadherin mAb clones SHE78-7 and HECD1 (M126 and M106; Takara), anti-vinculin (V9264; Sigma-Aldrich), anti-GFP (for Western) anti-afadin (sc-9996 and sc-74433; Santa Cruz Biotechnology), and anti-p120 (610134; BD Transduction Laboratories); Chicken anti-GFP (NB100-1614; Novus); Rabbit anti-p120 and anti-α-catenin (ab92514 and ab51032; Abcam, correspondingly), anti-Dsg2 (21880-I-AP; Proteintech), anti-mCherry (5993-100; BioVision), anti-PLEKHA5 (PA5-57463; Invitrogen); Guinea pig anti-pkp4 and anti-ARVCF (GP71 and GP155; Progen, correspondingly); and Sheep anti-erbin (AF7866; R&D Systems). Donkey Alexa Fluor 488-conjugated anti-rabbit (711-545-152), anti-mouse (715-545-150), anti-goat (705-545-147), and anti-guinea pig (706-545-148); donkey Cy3-conjugated anti-rabbit (711-165-152), anti-mouse (715-165-150), anti-goat (705-165-147), and anti-guinea pig (706-165-148); Alexa Fluor 647-conjugated donkey anti-guinea pig (706-605-148); and donkey horseradish peroxidase-conjugated anti-mouse (715-035-150), anti-rabbit (711-035-152), and anti-guinea pig (706-035-148) secondary antibodies were purchased from Jackson ImmunoResearch Laboratories.

### Live-cell imaging

The live-cell imaging experiments were performed essentially as described previously (Indra et al., 2018) using an X-Cite 120LED Boost High-Power LED Illumination System as a light source. In brief, cells were imaged in L-15 media with 10% FBS by an Eclipse Ti-E microscope (Nikon) at RT or 37°C controlled with Nikon's NIS-Elements software. The microscope was equipped with an incubator chamber, a Back-illuminated sCMOS Prime-95B camera (Photometrics), and a Plan-Apo TIRF 100×/1.49 lens. No binning mode was used in all live-imaging experiments. At this microscope setting, the pixel size was 110 nm. To monitor the entire lateral membrane from its basal to the apical edges, we obtained stacks of 5 focal planes with 0.5-μm spacing at each time point. All z-stack images were saved in ND2 format. The maximum intensity projection of each frame was created using

NIS Element 5.02. For movie analyses, all images were saved as Tiff files and processed using ImageJ software (National Institutes of Health). Lateral AJ motion was tracked using Fiji's TrackMate plugin with a DoG detector (object diameter: 1 µm, quality threshold: 1, auto thresholding). The Simple LAP tracker was set to a maximum linking distance and gap-closing distance of 1 µm and a max frame gap of 1. Tracks were color-coded according to time points of each frame. The plugin generated TrackMate XML files for the selected cell–cell contacts regions ($n$ = 3) containing only lateral AJs taken from three independent movies were used for MSD analysis. MSD plots were generated using R-studio with the TrackMateR package and were nearly identical for all three movies. MSD data can distinguish diffusive events, which fit a straight line, versus active motion, which fit a parabola.

FRAP was calculated using Fiji. Initially, several spot-like junctions were selected and precisely bleached with 405 lasers, and recovery was documented in every 10 s for a total of 500 s (for EcGFP, Fig 4 g) or in every 1 s for 100 s (for GFPp120 and GFPpkp4, Fig. 5 h).

### PREM
Sample preparation for immunogold PREM was performed as described previously (Korobova and Svitkina, 2008; Svitkina, 2009). In brief, cells grown on glass coverslips were sequentially extracted for 5 min with cytoskeleton preservation buffer (CPB, 100 mM PIPES, pH 6.9, 1 mM EGTA, and 1 mM $MgCl_2$) supplemented with 1% Triton X-100, then washed with CPB for 1 min, and incubated for 10 min with the anti-mCH antibody in CBP. Then the cells were washed for 5 min in CBP and incubated for an additional 10 min with Biotin-SP-VHH fragment of the Alpaca anti-rabbit antibody (611-065-215; Jackson ImmunoResearch Laboratories). The staining solutions were supplemented with 1% of Blocking Solution for Gold conjugates (Aurion). After another washing, the cells were fixed with 0.2% glutaraldehyde (for 10 min), quenched with 2 mg/ml NaBH4 in PBS (or 10 min), and stained with 10-nm gold-conjugated Streptavidin (ab270041; Abcam). After final washing (5 min), cells were postfixed with 2% glutaraldehyde in 0.1 M Na-cacodylate, pH 7.3. Fixed cells were treated with 0.1% tannic acid and 0.2% uranyl acetate, critical-point dried, coated with platinum and carbon, and transferred onto EM grids for observation. A parallel coverslip (stained as indicated above together with the control one without the anti-mCH antibody) was fixed in 3% formaldehyde and stained with mouse anti-Streptavidin antibody to verify staining specificity. EM samples were imaged using an FEI Tecnai Spirit G2 transmission electron microscope (FEI Company) operated at 80 kV. Images were captured by Eagle 4 k HR 200 kV CCD camera and presented in inverted contrast.

### Data processing
Chemiluminescence was detected via the Azure C300 Chemiluminescent Imager (Azure Biosystems), and band intensities were analyzed using ImageJ software (https://rsb.info.nih.gov/ij/). Most cell images were processed and analyzed using Nikon's NIS-Elements ver. 5.02. For line scan analysis, the Element's in-

built line profile function was used to draw a 1-pix-wide line across the junctions. For peak intensity measurement, the highest intensity on the y axis was recorded. A minimum of 15 independent junctions were scanned from five different images. To assess the ratio of the heterochromatic E-cadherin/specific protein-positive pixels versus the total number of the E-cadherin-positive pixels (Fig. 4, c and f), channels were split and treated with median filter of 1 pixel using Fiji. Both channels were autothresholded using Otsu algorithm. Thresholds were selected and added to the ROI manager. "AND" function was used to include the overlapping area from both channels. Percentage of overlap was calculated from the resulting analysis. For Pearson's correlation quantification, the arbitrary selected representative cell–cell contact areas were processed with the limited background reduction and denoising function of NIS-Element 5.02, and then Element's colocalization tool was used to obtain the scatterplots of red and green pixel intensities and to evaluate the Pearson's correlation coefficient (PCC). PCC of representative five cell–cell contact areas taken from three independent images were quantified for each staining (Fig. 2 e, Fig. S2, and Fig. S3). As an alternative approach to assess colocalization, we used the Colocalization Colormap plugin for ImageJ, which generates a pseudo-color map of correlations expressed in "normalized mean deviation product" (nMDP). The nMDP value reflects the correlation between the intensities of red and green pixels, ranging from –1 to +1. This plugin also generates an index of correlation, representing the fraction of positively correlated (colocalized) pixels. An example of this analysis is provided in Fig. S1 b, illustrating the spatial correlation of E-cadherin, pkp4, and p120 in the image presented in Fig. 1 a. Because the Colocalization Colormap approach yielded results essentially identical to those obtained using Pearson's correlation, which is more widely used, we did not present it to all images. The charts and error bars were plotted using GraphPad Prism version 10.2.0. Statistical significance was analyzed using student's two-tailed $t$ test for two groups. A P value that was <0.05 was considered statistically significant.

### Online supplemental material
Fig. S1 shows results related to Fig. 1 (the ratio of pkp4 to E-cadherin in junctions of different location, the colocalization color map to Fig. 1 b, and double staining for pkp4 and desmoglein 2) as well as results related to Fig. 3 (MSD analyses of lateral AJs). Fig. S2 shows results related to Fig. 4 (colocalization between GFP-tagged E-cadherin and pkp4, p120, or vinculin in cells deficient for pkp4 and p120). Fig. S3 shows results related to Fig. 5 (colocalization of GFPpkp4 and GFPp120 with endogenous E-cadherin). Fig. S4 shows results related to Fig. 6 c (colocalization of αCatCH-Δ259 with vinculin, erbin, and PLEKHA5). Fig. S5 shows results related to Fig. 7 b (colocalization of pkp4 with desmoglein 2, E-cadherin, and F-actin in cells deficient for α-catenin and plakophilins 2 and 3).

## Data availability
The data underlying all figures in this manuscript are available in the article and its online supplemental materials. Additionally,

any original data including microscope or western blot images generated during this study are available upon request.

## Acknowledgments

We thank Dr. T. Svitkina (University of Pennsylvania) for valuable comments and suggestions. Sequencing, flow cytometry, and confocal microscopy were performed at the Northwestern University Genetic, Flow Cytometry, and Advanced Microscopy Centers.

The work was supported by National Institute of Health Grants AR070166 and GM148571 (to S.M. Troyanovsky).

Author contributions: Indrajyoti Indra: data curation, formal analysis, funding acquisition, investigation, methodology, software, supervision, validation, visualization, and writing—original draft. Regina B. Troyanovsky: formal analysis, investigation, methodology, project administration, resources, and writing—original draft, review, and editing. Farida V. Korobova: investigation and writing—review and editing. Sergey M. Troyanovsky: conceptualization, data curation, formal analysis, funding acquisition, investigation, methodology, project administration, supervision, validation, visualization, and writing—original draft, review, and editing.

Disclosures: The authors declare no competing interests exist.

Submitted: 18 July 2025

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

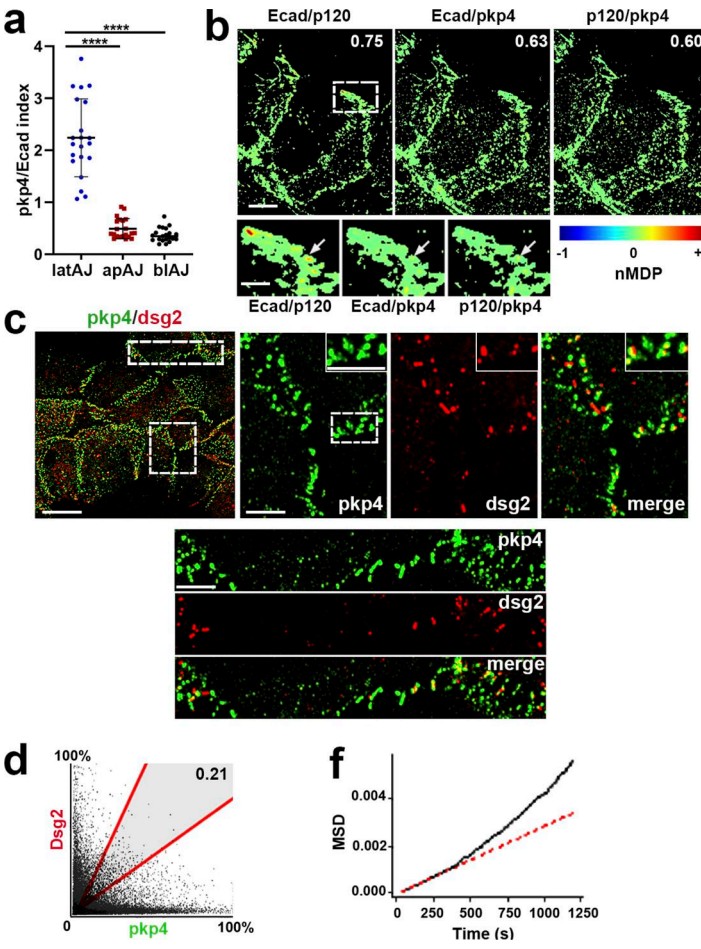

Figure S1. **Pkp4 is enriched at lateral AJs. (a)** Ratio of pkp4 to E-cadherin fluorescence intensities (pkp4/Ecad index) assessed for lateral, apical, and basal AJs (20 junctions each) across four cell–cell contact regions from two independent images. Pkp4 levels are significantly higher in lateral AJs compared with apical or basal AJs. Data are presented as mean ± SD. Statistical significance was determined using a two-tailed Student's $t$ test, ****P <0.0001. **(b)** Colocalization color maps depicting colocalization between E-cadherin-p120, E-cadherin-pkp4, and p120-pkp4. The −1 to +1 heat map displays the nMDP, which represents correlation between intensities of corresponding pixels: negative correlation (-1-0) are shown in blue-green colors, whereas positive correlations (0 to +1) are represented by yellow-red colors. The index of correlation (Icorr) for each protein pair is shown in the upper right corner. Bar, 20 µm. Consistent with Pearson's analyzes (Fig. 1), this approach reveals the strongest correlation between E-cadherin and p120, and the weakest between p120 and pkp4. It also shows that apical AJs contain isolated pkp4-enriched clusters (one example is indicated by an arrow in the zoomed images at the bottom panel). Bar, 10 µm. **(c)** Maximum-intensity projections of A431 cells stained for pkp4 (green) and for desmosomal protein dsg2 (dsg2, red). Only the merged image is shown at low magnification. Bar, 20 µm. Zoomed regions (dashed boxes) are shown at the bottom panel and in the inset for each channel separately. Bar, 10 µm. Note that DSMs do not colocalize with pkp4-rich AJs, although they are frequently adjacent. **(d)** Scatterplot of red versus green relative pixel intensities for the image in panel c. The PCC (r) is indicated in the upper right corner. Note the nearly complete absence of positive correlation. **(f)** MSD measured from a region containing ∼16 lateral AJs from Video 1 (black trace). Note that the observed MSD deviates from a linear trend (red line) suggesting the active, non-Brownian movement of the junctions.

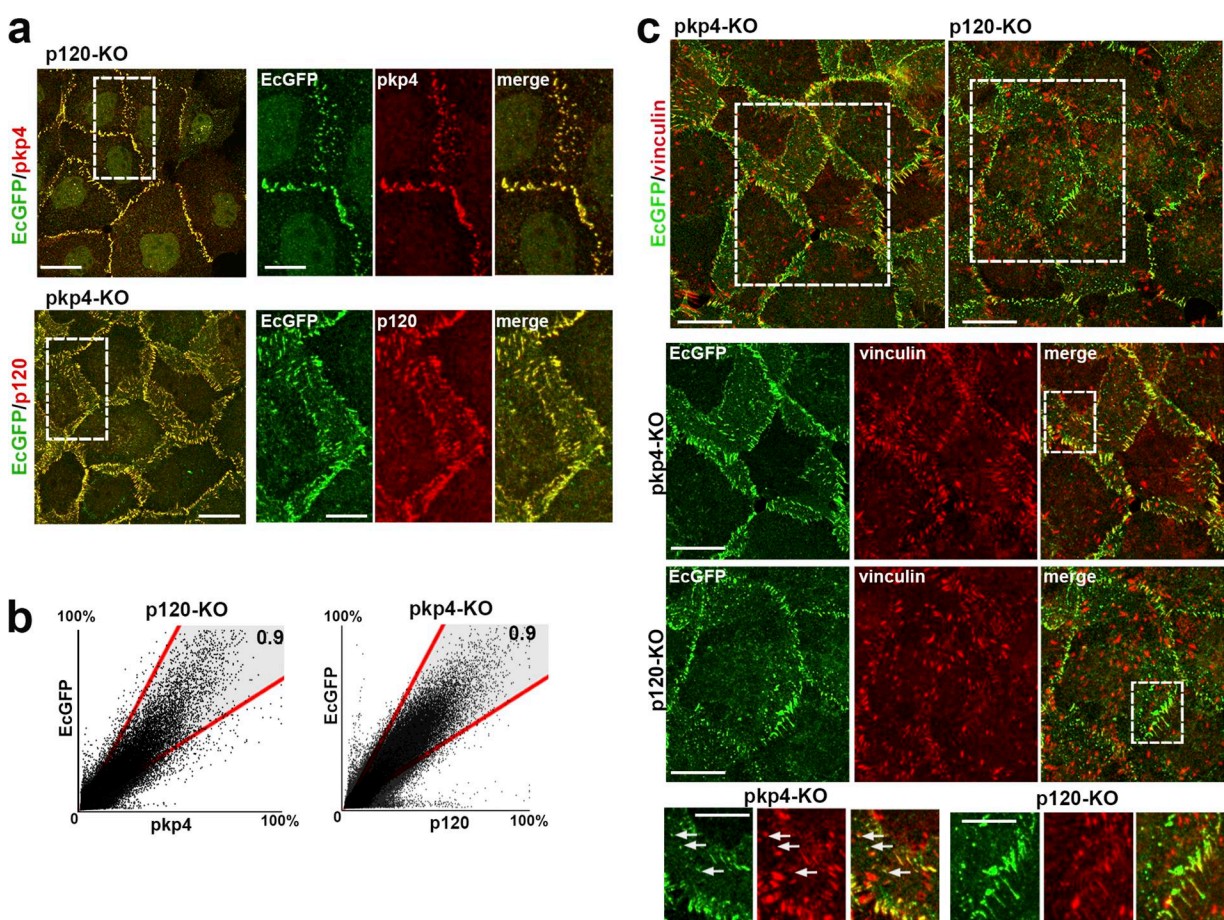

Figure S2. **Phenotype of EcGFP-A431 cells deficient for pkp4 and p120. (a)** Maximum-intensity projections of all x–y optical slices of the p120-KO and pkp4-KO EcGFP-A431 cells stained for EcGFP (GFP, green) and pkp4 or p120 (both red), respectively. Only merged images are shown for low magnifications. Bar, 20 µm. The zoomed areas (marked by dashed boxes) are shown at right in both colors. Bar, 10 µm. **(b)** The scatterplots of red-green relative pixel intensities of the images shown in a. The Pearson's r values are in the upper right corners. Note that the cells show nearly perfect positive relationship between the fluorophores. **(c)** Projections of all x–y slices of the same knockout cells stained for EcGFP (green) and vinculin (red). Only merged images are shown at low magnification. Bar, 20 µm. Enlarged regions (dashed boxes) are presented below in both channels. Bar, 15 µm. Its dashed box area is further magnified in the bottom row. Bar, 8 µm. Arrows indicate vinculin-positive lateral AJs observed in pkp4-KO cells.

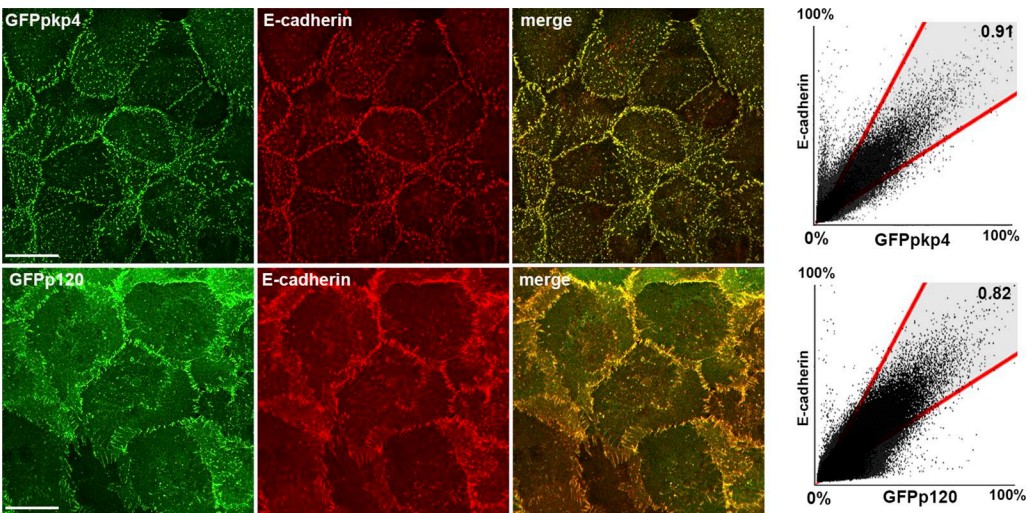

Figure S3.   **GFPpkp4 and GFPp120 co-localize with E-cadherin in δCat-KO cells.** Maximum-intensity projections of all x–y optical slices of GFPpkp4- and GFPp120-expressing δCat-KO cells lacking endogenous p120, pkp4, and ARVCF, stained for GFP (green) and E-cadherin (red). Bar, 20 µm. Corresponding scatterplots of green versus red relative pixel intensities are shown at right panel, with r values indicated in the upper right corners. Note that both GFPpkp4 and GFPp120 display an almost perfect positive correlation with E-cadherin.

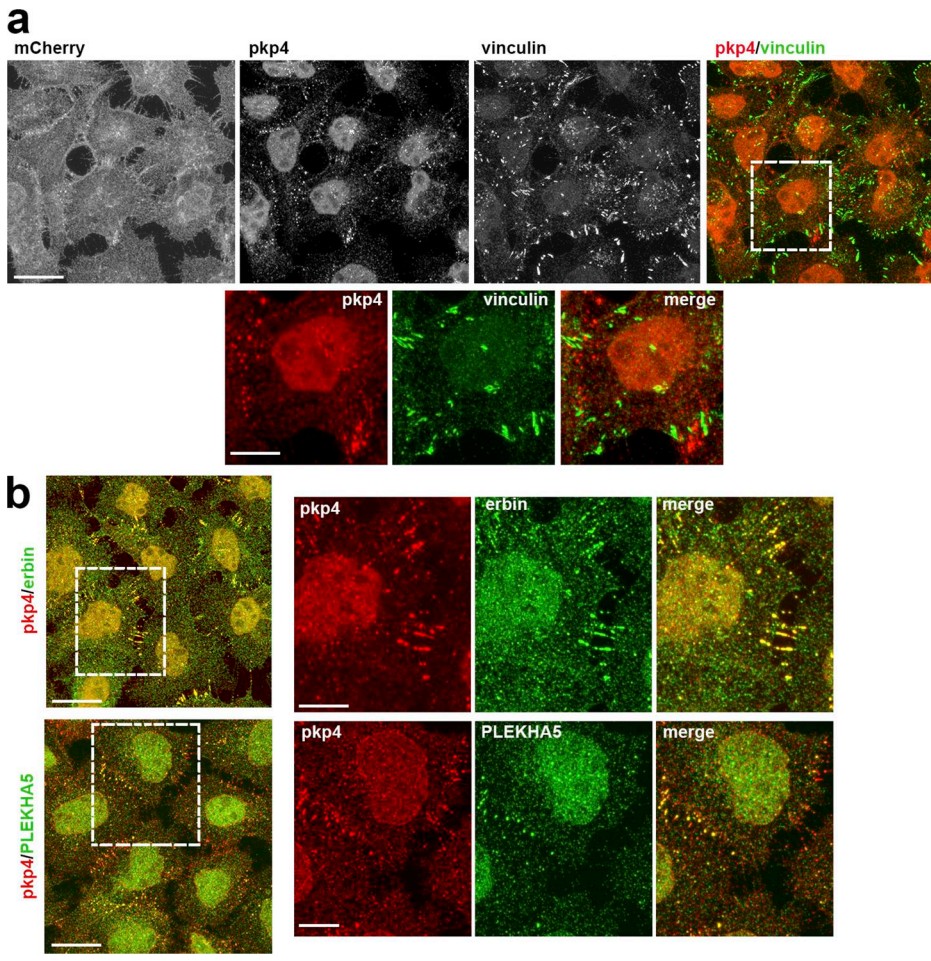

Figure S4. **GFPpkp4-E-cadherin clusters in α-catenin–deficient cells display characteristics of lateral AJs. (a)** Confocal fluorescence microscopy (maximum-intensity projections of all x–y slices) of αCat-KO cells expressing αCatCH-Δ259 triple stained for mCherry, pkp4, and vinculin. Bar, 20 µm. Detailed subcellular distribution of pkp4 (red) and vinculin (green) within the regions marked by white dashed boxes are shown below. Bar, 10 µm. **(b)** Maximum-intensity projections of all x–y optical slices of the same cells double-stained for pkp4 (red) and erbin or PLEKHA5 (green). Only merged images are shown at low magnifications. Bar, 20 µm. Enlarged areas (indicated by white dashed boxes) are shown at right for both channels. Bar, 10 µm. αCat, α-catenin; mCH, mCherry tag.

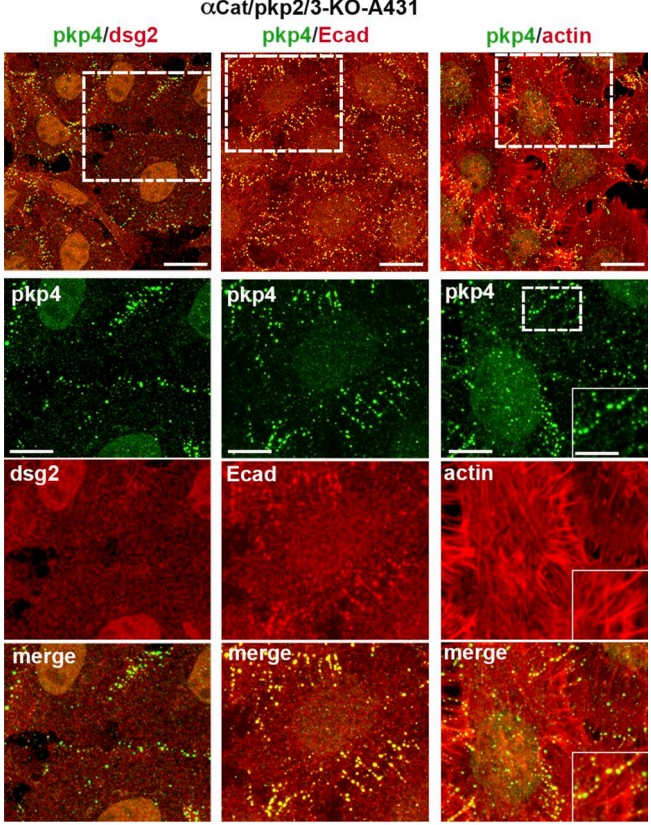

Figure S5. **DSMs are not involved in pkp4-facilitated E-cadherin clustering.** Maximum-intensity projections of all x–y optical slices of αCat/pkp2/3-KO cells stained for pkp4 (green) and co-stained for the desmosomal marker, dsg2, or E-cadherin (Ecad), or F-actin (all in red). Only merged images are shown at low magnifications. Bar, 20 µm. Enlarged regions (indicated by white dashed boxes) are shown for both channels at the bottom. Bar, 10 µm. The boxed region within the enlarged F-actin image is further magnified in the insets (Bar, 7.5 µm). Note that cells form pkp4 clusters despite the complete absence of DSMs and α-catenin. Also note that the vast majority of pkp4 clusters are associated with actin-rich structures. αCat, α-catenin.

Video 1.   **Dynamics of the AJs at the selected cell–cell contact of the control EcGFP-expressing A431 cells.** Time-lapse of EcGFP. The most representative cell–cell contact from six independent movies was selected. The images were acquired at 30-s intervals and are shown with display rate 10 frames/s. See Fig. 3 for details.

Video 2.   **Dynamics of the AJs at the selected cell–cell contact of the control EcGFP-expressing pkp4-KO cells.** Time-lapse of EcGFP. The most representative cell–cell contact from six independent movies was selected. The images were acquired at 30-s intervals and are shown with display rate 10 frames/s. See Fig. 4 d for details.

Video 3.   **Dynamics of the AJs at the selected cell–cell contact of the control EcGFP-expressing p120-KO cells.** Time-lapse of EcGFP. The most representative cell–cell contact from six independent movies was selected. The images were acquired at 30-s intervals and are shown with display rate 10 frames/s. See Fig. 4 e for details.

Video 4.   **Dynamics of the AJs at the selected cell–cell contact of the GFPpkp4-expressing δCat-KO cells.** Time-lapse of GFPpkp4. The most representative cell–cell contact from eight independent movies was selected. The images were acquired at 30-s intervals and are shown with display rate 10 frames/s. See Fig. 5 e for details.

Video 5.   **Dynamics of the AJs at the selected cell–cell contact of the GFPp120-expressing δCat-KO cells.** Time-lapse of GFPp120. The most representative cell–cell contact from eight independent movies was selected. The images were acquired at 30-s intervals and are shown with display rate 10 frames/sec. See Fig. 5 f for details.

