## [Peer Review File · The Journal of Cell Biology]

Two δ -Catenins, Plakophilin 4 and p120, Promote Formation of Distinct Types of Adherens Junctions

Indrajyoti Indra, Regina Troyanovsky, Farida Korobova, and Sergey Troyanovsky

Corresponding Author(s): Sergey Troyanovsky, Northwestern Medicine

Review Timeline:

Submission Date:	2025-07-18
Editorial Decision:	2025-09-08
Revision Received:	2025-11-14
Editorial Decision:	2025-12-15
Revision Received:	2025-12-17

Monitoring Editor: Ian Macara

Scientific Editor: Dan Simon

Transaction Report:

DOI: <https://doi.org/10.1083/jcb.202507134>

September 8, 2025

Re: JCB manuscript #202507134

Sergey Troyanovsky
Northwestern Medicine

Dear Dr. Troyanovsky,

Thank you for submitting your manuscript entitled "Two d-Catenins, Plakophilin 4 and p120, Promote Formation of Distinct Types of Adherens Junctions." The manuscript was assessed by expert reviewers, whose comments are appended to this letter. We invite you to submit a revision as an Article if you can address the reviewers' key concerns, as outlined here.

You will see that the reviewers feel that the identification of distinct types of AJs represents an important conceptual advance for the field. Reviewer #1 asks for minor changes to text and figures. Reviewer #2 requests orthogonal colocalization analyses, a more nuanced interpretation of the displacement data, to confirm that pkp4 directly interacts with F-actin, and to assess the localizations of pkp4-cadherin clusters relative to desmosomes in wildtype cells. All of these should be addressed in full. Reviewer #3 asks for more insight into the molecular mechanisms and functional roles of the pkp4-containing lateral AJs and provided several excellent suggestions for additional experiments. Of these we feel that functional studies into the roles of lateral AJs would be most important to add in revision and this should be the focus of your efforts. It is also important to test whether pkp4 is necessary for AJs clustering along F-actin in the absence of a-catenin's actin binding region and add quantifications of pkp4 enrichment at lateral AJs. While identifying additional pkp4-dependent AJs components and investigating the roles of extracellular regions of AJ components in clustering are interesting questions, we do not believe that these are essential for this paper.

GENERAL GUIDELINES:

Text limits: Character count for an Article is < 40,000, not including spaces. Count includes title page, abstract, introduction, results, discussion, and acknowledgments. Count does not include materials and methods, figure legends, references, tables, or supplemental legends.

Figures: Articles may have up to 10 main text figures. Figures must be prepared according to the policies outlined in our Instructions to Authors, under Data Presentation, <https://jcb.rupress.org/site/misc/ifora.xhtml>. All figures in accepted manuscripts will be screened prior to publication.

*****IMPORTANT:** It is JCB policy that if requested, original data images must be made available. Failure to provide original images upon request will result in unavoidable delays in publication. Please ensure that you have access to all original microscopy and blot data images before submitting your revision. ***

Supplemental information: There are strict limits on the allowable amount of supplemental data. Articles may have up to 5 supplemental figures. Up to 10 supplemental videos or flash animations are allowed. A summary of all supplemental material should appear at the end of the Materials and methods section.

Please note that JCB now requires authors to submit Source Data used to generate figures containing gels and Western blots with all revised manuscripts. This Source Data consists of fully uncropped and unprocessed images for each gel/blot displayed in the main and supplemental figures. For assays performed using capillary electrophoresis and/or immunoassay-based detection, authors should instead provide the electropherogram graph(s) for each experiment, plotting fluorescence/chemiluminescence intensity vs. molecular weight/size. Please be sure to provide one Source Data file for each figure gels, blots, and/or capillary electrophoresis assays along with your revised manuscript files. File names for Source Data figures should be alphanumeric without any spaces or special characters (i.e., SourceDataF#, where F# refers to the associated main figure number or SourceDataFS# for those associated with Supplementary figures). For traditional gels and blots, the lanes of the gels/blots should be labeled as they are in the associated figure, the place where cropping was applied should be marked (with a box), and molecular weight/size standards should be labeled wherever possible. For capillary electrophoresis assays, each trace in the graph should be color-coded and labeled to indicate which protein, gene, or sample is being measured (please try to avoid red/green combinations to accommodate our color-blind readers).

The typical timeframe for revisions is three to four months. If you anticipate any difficulties in meeting this aforementioned revision time limit, please contact us and we can work with you to find an appropriate time frame for resubmission. Please note that papers are generally considered through only one revision cycle, so any revised manuscript will likely be either accepted or rejected.

Thank you for this interesting contribution to Journal of Cell Biology. You can contact us at the journal office with any questions at cellbio@rockefeller.edu.

Sincerely,

Ian Macara, PhD
Monitoring Editor
Journal of Cell Biology

Dan Simon, PhD
Scientific Editor
Journal of Cell Biology

Reviewer #1 (Comments to the Authors (Required)):

Although morphological diversity of adherens junctions has long been recognized, differences in molecular composition and function have not been examined in detail. The manuscript by Indra et al addresses this knowledge gap by identification and thorough characterization of adherens junction subtypes. They focus on delta-catenins, linker molecules that bind to the same cytoplasmic sites of the transmembrane E-cadherin adhesion receptors of adherens junctions. delta-catenins comprise three members, i.e. p120, pkp4 and ARVC, which are often co-expressed in the same cell. Assuming that they compete for binding to E-cadherin the authors examined their distribution in squamous cell carcinoma-derived A431 and adenocarcinoma-derived DLD1 cells. By limiting their detailed analyses to p120 and pkp4 they make the remarkable observation that each of them defines a distinct type of adherens junction: the "classical" p120-positive junctions include apical, immobile and basal, upward-moving punctate junctions that are positive for the tension-sensitive components vinculin and afadin, whereas novel type of pkp4-positive junction localizes to lateral membrane contact regions. These pkp4 junctions are oscillatory without directed motion, lack afadin and vinculin, but are, instead, positive for the signaling molecule PLEKHA5. FRAP analyses show that p120 junctions have a high turnover, whereas pkp4 junctions have a very low turnover. Elegant gain- and loss-of-function experiments corroborate the relevance of the two catenins for generating the different junctional entities. Further experiments reveal the importance of p120 junctions for monolayer contraction and adhesion in contrast to the negligible contribution of pkp4 junctions. Probably the most remarkable finding is that pkp4 junctions enable actin filament recruitment in an alpha-catenin-independent fashion and obviously without the contribution of either vinculin or afadin. Expressing an alpha-catenin rescue mutant lacking all possible actin-binding domains, they still observe actin-bound pkp4 clusters as well as clusters of actin filament bundle-associated mutant alpha-catenin accumulations using PREM immunogold-labeling.

The manuscript excels by providing conclusive and solid evidence for adherens junction diversification in a single cell type with beautiful images and thorough analyses at different levels employing high resolution immunolocalization, time-lapse imaging, FRAP, biochemical analysis and electron microscopy. The resulting conceptual advance will spur future lines of research to elucidate the molecular and functional wealth of adherens junctions and their connectivity.

I have only a few minor points to consider:

- I do not understand the difference between "spot-like" and "punctate". Different wording would help to avoid confusion and improve clarity.
- I could not find detailed information on the delineation of the apical versus lateral versus basal domains. Please specify!
- Annotations should be included in the videos by adding time points, scale bars and cell type.
- Lines 224/225: I do not understand the sentence. Check whether it is correct.
- Figure 6: Include an explanation for the N1/N2 domains of -catenin in the legend. The "red" pseudocolor of the immunogold particles is difficult to discern.

- Lines 518/519: I assume that the FRAP experiments in Figure 4g were performed on EcGFP. Please, clarify or correct.
- Line 765: "left" should read "right".
- Line 806: "bt" should probably read "bl".
- Line 810: What are "three independent images"?
- How do the junctions relate to the different actin networks observed in vivo, e.g. in polarized intestinal cells (Barai et al., Nature Communications (2025) 16:6201)?
- Although English is not my mother tongue, I believe that careful editing will improve the manuscript.

Reviewer #2 (Comments to the Authors (Required)):

In this manuscript the Troyanovsky lab describes a previously un-anticipated formation of cadherin complex that does not apparently connect to the cytoskeleton via alpha-catenin. Although, as reported by the authors, many previous and even old papers reported diversity in adherens junction ultrastructure, composition, morphology and dynamics in different cell types and even within epithelial cells, no clear data explained the coexistence of different clusters of cadherin within a same cell. Here the authors show that two members of the delta-catenin protein family, p120 and plakophilin 4 (pkp4), which interact with the juxtamembrane intracellular region of classic cadherins, promote distinct types of cadherin clustering thereby contributing to AJ specialization. The type controlled by p120 is driven by interactions between cadherin-associated protein, α -catenin, and actin filaments. This "canonical" clustering mechanism results in formation of AJs that play a major role in overall cell-cell adhesion. The type promoted by pkp4 is driven by an α -catenin-independent cadherin-F-actin interaction. It generates the so-called lateral spot AJs.

The manuscript overall presents very nice and convincing imaging data in WT cells and cells with both loss of function and gain of function of each of these two relatives. The data support most of the conclusions presented by the authors without however digging in the mechanisms by which the novel complex associated to pkp4 associates to actin, or in addition to other cytoskeleton components. This limits a bit the impact of the study but a few well-designed experiments could very quickly address the two main hypotheses presented by the authors at the end of the manuscript (see below). Another obvious hypothesis that is suggested by the authors is the possible interconversion between the two type of clusters. The authors have previously extensively used photoconversion of Dendra and such analysis should quickly test this hypothesis.

Specific comments:

The analysis of colocalizations in Figures 1, 2, S1, S2 which is a major metrics to demonstrate and quantify the existence of the different clusters based only on Pearson analysis. Colocalization analysis is a tricky analysis and had often demonstrated some limitation, it is the case for the Pearson. Other authors used other methods, based for example on image segmentation and more complex ones. It would be nice if the authors could double their colocalization analysis by another one. Specially because depending of the couples of molecules analyzed, the colocalization is mild.

When authors analyze the displacement of the spots (clusters), they should also try more powerful tools as MSD analysis. Line 164-165, they say "lateral sAJs exhibited oscillatory motion". I think this is not a good term. MSD analysis should tell whether the displacements are Brownian, directed motion, etc... or combination.

Line 170-177 and other places in the manuscript, authors combine in their description of the results what they see in their data and what is known for these spots in literature which is misleading. Example line 174-176: authors do not clearly show and analyze the full movement of cluster from their formation to merge to apical AJ.

I am not convinced by the conclusion of Figure 7 experiments saying that the pkp4 complexes are not involved in cell-cell adhesion. The absence of contribution to cell-cell adhesion could in fact merely reflect the relative abundance of these complexes compare to the p120 complexes. I think that the claim is inappropriate.

To come back to the mechanisms leading to the linkage of these new clusters to the cytoskeleton, the authors propose the direct interaction of pkp4 with F-actin via a previously identified region of the molecule. The authors have almost in hands all the tools (double pkp KO cells and pkp4 DNA) to test this quickly and they should do it.

For a relation with desmosomes, this is only addressed by *dsg2* staining in pkp 2/3 mutants. They should at least look at *dsg2* colocalization with the cadherin complexes already in wt cells (Figure 1): how distributes the pkp4-cadherin clusters compared to desmosomes.

Reviewer #3 (Comments to the Authors (Required)):

In the manuscript "Two δ -Catenins, Plakophilin 4 and 120, Promote Formation of Distinct Types of Adherens Junctions" Indra et al. investigate the mechanisms underlying compositional and functional diversity in adherens junction (AJ) complexes. They demonstrate convincingly that two populations of E-cadherin-containing AJs are distinguished / specified by which δ -catenin they contain. Adhesions featuring p120 also contain reinforcing proteins canonically associated with mechanotransduction (vinculin and afadin) and are linked to the actin cytoskeleton through alpha-catenin. Conversely, adhesions featuring pkp4 are smaller, localized to a distinct subcellular region, lack reinforcing proteins, and appear to be strongly coupled to the actin cytoskeleton through an alternative, alpha-catenin independent mechanism.

The experimental data presented are convincing, and the identification of molecular constituents which mediate AJ functional diversity are a significant discovery which represent an important advance for the field. However, I believe the level of mechanistic insight presented (specifically, the lack of data demonstrating how these constituents exert their functional effects) is below that I would expect for an Article in JCB. It could be worth considering condensing the paper into a Report, a format where exciting discoveries with less mechanistic support are appropriate. I detail below the specific mechanistic / functional weaknesses, which admittedly would require a level of effort to address beyond the standard expectations of a revision, as well as several more minor issues.

Major mechanistic / functional weakness:

1) Mechanism of cytoskeleton coupling through pkp4

The authors convincingly show alpha-catenin independent coupling between pkp4 containing adhesions and the actin-myosin cytoskeleton, an exciting observation. However, they do not identify the alternative coupling mechanism. Doing so would substantially enhance the level of advance offered by the paper. This could be done e.g. by identifying additional pkp4-dependent constituents of these AJs and probing their role in cytoskeletal coupling.

2) Function of pkp4-containing lateral AJs

The authors also show that the pkp4-containing AJs that they identify are not necessary for cell adhesion by a standard assay, but the functional relevance of this class of adhesion is not further elucidated. The authors speculate that "dramatic effects observed in p120-KO cells were caused by the reduced level of E-cadherin but not by general inability of pkp4-CCC to maintain cell-cell adhesion" (lines 337-339). However, they do not provide any evidence for what these adhesions actually do. For example, do these lateral clusters interact with those in the neighboring cells, and do they play any role in the maturation of epithelial barriers? Adding such functional insights would also substantially strengthen the paper.

Minor points:

4. The authors show that AJs can cluster along F-actin in the absence of a-catenin's actin binding region, but they do not show that pkp4 is required for this behavior. These AJ clusters are referred to as being generated by pkp4-CCC, but the authors don't show that pkp4 is present in the Triton X-100 resistant structures.

5. The authors state that AJ clusters are formed by oligomerization of components inside the cell, but they do not assess how the extra-cellular regions of, e.g. E-cadherin, contribute to the clustering behavior of the different AJ classes.

6. The authors claim that "pkp4 appeared to be most prominent in AJs located at the middle portion of the lateral membrane" (lines 122-123), which is supported by a single image in Fig. 1c. This claim should be supported with quantification.

Textual / presentation issues:

7. The paper would benefit from having the fluorescent images being presented in magenta / green, rather than red / green, for colorblind readers.

8. Figure 4 (b) legend states that "The arrows point at few remaining PLEKHA5-positive AJs in p120-KO cells.", but it appears that it is referring to pkp4-KO cells.

General notes

We are grateful to all reviewers for their constructive comments and for recognizing our results as an important conceptual advance in the field. We have made every effort to address all points raised. In particular, as suggested by Reviewer #2, we added new quantifications of E-cadherin colocalization with p120 and pkp4, as well as MSD analyses of lateral AJ motility (Fig. S1b, f). We also performed additional experiments in response to comments from Reviewers #2 and #3, including analyses of the relationship between lateral AJs and desmosomes (Fig. S1c), trans-interactions within lateral AJs (Fig. 7c), and the solubility of Pkp4 following Triton X-100 extraction (Fig. 6f). All major revisions addressing these points are highlighted in red. We believe that the new data and textual changes make the manuscript more logical and straightforward.

Unfortunately, two issues raised by the reviewers could not be fully addressed at this stage. Reviewers #2 and #3 requested mechanistic insights into how lateral AJs interact with actin. While we fully agree that this is an important question, it lies beyond the scope of the present study. Elucidating the principles of cadherin-actin coupling through α -catenin required many years and the efforts of multiple laboratories. Our preliminary work indicates that the mechanisms mediating cadherin-actin interactions in lateral AJs are at least as complex. We have begun to explore these mechanisms and hope to report our findings in the near future.

The second unresolved question concerns the specific physiological functions of lateral AJs. Addressing this will require extensive genetic, proteomic, and in situ studies of cells within native tissues. We have cited available data suggesting that lateral AJs may contribute to signaling. We hope that our findings, by defining the distinct nature of these junctions, will stimulate and guide future research in this direction.

Reviewer #1

We thank this reviewer for the conclusion that *“The manuscript excels by providing conclusive and solid evidence for adherens junction diversification in a single cell type with beautiful images and thorough analyses at different levels employing high resolution immunolocalization, time-lapse imaging, FRAP, biochemical analysis and electron microscopy. The resulting conceptual advance will spur future lines of research to elucidate the molecular and functional wealth of adherens junctions and their connectivity.”*

Several minor concerns raised by this reviewer are addressed below:

1) I do not understand the difference between "spot-like" and "punctate". Different wording would help to avoid confusion and improve clarity.

Thank you for pointing out the ambiguity in the definition of adherens junctions (AJs). We fully agree that the current classification of AJs is confusing, primarily because it has developed from numerous independent and often context-specific observations by different researchers. However, as we summarized in the Introduction, this classification emphasizes the high degree of structural flexibility of AJs and variability of their interactions with the cytoskeletal. Indeed,

even within a single cell, AJ morphology can be remarkably diverse and dynamically changing. In the revised version of the manuscript, we have simplified our terminology related to AJs. Because the most relevant feature of AJs for our study is their localization, we have removed the terms “spot-like” and “punctate,” retaining only localization-based portion of the term: apical, lateral, or basal.

2) I could not find detailed information on the delineation of the apical versus lateral versus basal domains. Please specify!

This delineation is, of course, arbitrary (as we now note on line 194), although we believe it likely reflects underlying differences in the actin cortex. However, we prefer not to engage in such speculation, as it lies beyond the scope of our study.

3) Annotations should be included in the videos by adding time points, scale bars and cell type.

All requested information is now provided.

4) Lines 224/225: I do not understand the sentence. Check whether it is correct.

The sentence is clarified.

5) Figure 6: Include an explanation for the N1/N2 domains of α -catenin in the legend. The "red" pseudocolor of the immunogold particles is difficult to discern.

Because the subdomain organization of the α -catenin N-terminal domain is not relevant to our study, this detail has been removed from the figure. The functional role of the N-terminal domain has been added to the legend. The red pseudocolor in Figure 6g has been intensified to improve clarity.

6) Lines 518/519: I assume that the FRAP experiments in Figure 4g were performed on EcGFP. Please, clarify or correct.

Clarifications are added (line 538-539).

6) Line 765: "left" should read "right".

Corrected.

7) Line 806: "bt" should probably read "bl".

Thank you for noting.

8) Linie 810: What are "three independent images"?

We clarified the corresponding quantifications in the M&M section (lines 574-575).

9) *How do the junctions relate to the different actin networks observed in vivo, e.g. in polarized intestinal cells (Barai et al., Nature Communications (2025) 16:6201)?*

We did not observe actin cytoskeletal structures resembling the basally located “actin stars” described by Barai et al. in either of our cell models, A431 or DLD1. However, the specific features of basal and lateral AJs clearly indicate a specialized organization of the actin cortex within the corresponding regions of the lateral membrane domain. A reference to the study by Barai et al. has been added to the Introduction.

10) *Although English is not my mother tongue, I believe that careful editing will improve the manuscript.*

We hope that the extensive editing addresses this general complain.

Reviewer #2

Summarizing our manuscript this reviewer noted that *“The manuscript overall presents very nice and convincing imaging data in WT cells and cells with both loss of function and gain of function of each of these two relatives. The data support most of the conclusions presented by the authors without however digging in the mechanisms by which the novel complex associated to pkp4 associates to actin, or in addition to other cytoskeleton components.”*

He/she then added that a lack of mechanistic details of how lateral AJs interact with actin filaments *“limits a bit the impact of the study but a few well-designed experiments could very quickly address the two main hypotheses presented by the authors at the end of the manuscript (see below). Another obvious hypothesis that is suggested by the authors is the possible interconversion between the two type of clusters. The authors have previously extensively used photoconversion of Dendra and such analysis should quickly test this hypothesis.”*

Unfortunately, it is not feasible to analyze the interconversion of basal AJs into lateral AJs using photoconversion. Photoconversion of a basal AJ generates a pool of “red” E-cadherin that gradually redistributes into all AJs surrounding the photoconverted region. Similar experiments previously conducted in our laboratory (Hong et al., 2010, doi: 10.1073/pnas.0911027107) demonstrated that AJs, while remaining morphologically intact, undergo continuous molecular turnover. Therefore, it remains entirely possible that lateral AJs incorporate E-cadherin originating from disassembled basal AJs. To clarify this point, we have revised the corresponding section of the manuscript. In addition, we carefully re-examined our live-cell imaging data to identify clear examples of basal-to-lateral AJ transitions. One such potential event is presented below; however, even in this case, definitive interpretation is not possible. Thus, the assembly and potential interconversion of lateral AJs require a dedicated investigation. These points are now emphasized in the revised text (lines 190-193): *“Occasionally, basal AJs that initially moved upward transitioned into a more nondirectional, lateral AJ-like motion. Whether these rare events represent interconversion of basal AJs into lateral AJs, and if so, what mechanisms might underlie such interconversion, remains unclear.”*

A rare example of an apparent interconversion of a basal AJ into a lateral AJ. The junction, initially located at the base of the lateral membrane at the beginning of the movie (shown in the image), moved persistently upward until it reached the region containing lateral AJs, where its movement markedly slowed.

Specific comments:

1). The analysis of colocalizations in Figures 1, 2, S1, S2 which is a major metrics to demonstrate and quantify the existence of the different clusters based only on Pearson analysis. Colocalization analysis is a tricky analysis and had often demonstrated some limitation, it is the case for the Pearson. Other authors used other methods, based for example on image segmentation and more complex ones. It would be nice if the authors could double their colocalization analysis by another one. Specially because depending of the couples of molecules analyzed, the colocalization is mild.

To test the robustness of the PCC values shown in Figs. 1, 2, S1, and S2, we recalculated them using Costes' automatic thresholding algorithm implemented in the JACoP plugin for ImageJ. The PCC values obtained with this method were nearly identical to those calculated in NIS-Elements 5.02. Moreover, the plugin returned Costes p-values of +1 for all analyzed images, confirming the statistical reliability of the correlations.

As an additional approach, we employed the Colocalization Colormap plugin for ImageJ, which produces pseudo-color maps based on the "normalized mean deviation product" (nMDP), a pixel-by-pixel correlation coefficient ranging from -1 to $+1$. This analysis also provides an index of correlation (Icorr), reflecting the fraction of positively correlated (colocalized) pixels. An example of this analysis is now presented in Fig. S1b, illustrating the spatial correlation among E-cadherin, pkp4, and p120 in the image shown in Fig. 1a. Because the nMDP approach yielded results essentially identical to those obtained using Pearson's correlation, which is more widely used, we did not present these results to all images in the study, but indicated the use of both methods in the Materials and Methods section.

2). When authors analyze the displacement of the spots (clusters), they should also try more powerful tools as MSD analysis. Line 164-165, they say "lateral sAJs exhibited oscillatory motion". I think this is not a good term. MSD analysis should tell whether the displacements are Brownian, directed motion, etc... or combination.

We used the term "oscillatory motion" because it has been used previously to describe the dynamics of lateral AJs (Wu et al., 2014, doi: 10.1016/j.ejcb.2014.09.001). As suggested, we performed mean square displacement (MSD) analyses of our time-lapse movies using R-Studio with the TrackMateR package, which confirmed that the lateral AJs exhibit active motion (Fig. S1f). This observation is consistent with the published data showing their interaction with the actin cytoskeleton and implication of myosin II in their movement (Wu et al., 2014).

3). Line 170-177 and other places in the manuscript, authors combine in their description of the results what they see in their data and what is known for these spots in literature which is misleading. Example line 174-176: authors do not clearly show and analyze the full movement of cluster from their formation to merge to apical AJ.

We tried to identify the places indicated by the reviewer and clarified how our observations align with published data. For example, we now specify (lines 185-186) that “*Consistent with published data (Hong et al., 2010; Kametani & Takeichi, 2007), basal AJs were continuously generated at the basal end of the lateral plasma membrane and then moved upward (green tracks in Fig 3c).*”

4). I am not convinced by the conclusion of Figure 7 experiments saying that the pkp4 complexes are not involved in cell-cell adhesion. The absence of contribution to cell-cell adhesion could in fact merely reflect the relative abundance of these complexes compare to the p120 complexes. I think that the claim is inappropriate.

We have clarified our interpretation of the role of the pkp4-CCC in adhesion. This protein complex does mediate adhesion in the absence of p120, forming AJs through the α -catenin-dependent mechanism. We observed such AJs by vinculin staining in p120-KO cells and in δ Cat-KO cells expressing GFPpkp4 (Figs. 5 and S3). In contrast, we did not detect any measurable adhesion between cells connected exclusively by α -catenin-independent AJs, which are formed in α Cat/pkp2/3-KO cells and in their GFPpkp4-expressing derivatives, even though these cells generated numerous dot-like AJs. We incorporated this clarification into the revised manuscript (lines 461-462) and added a new experiment (as requested by Reviewer #3) demonstrating that AJs produced in α -catenin-KO cells contain cadherin clusters engaged in trans interactions (Fig. 7c).

5). To come back to the mechanisms leading to the linkage of these new clusters to the cytoskeleton, the authors propose the direct interaction of pkp4 with F-actin via a previously identified region of the molecule. The authors have almost in hands all the tools (double pkp KO cells and pkp4 DNA) to test this quickly and they should do it.

A similar question was raised by Reviewer #3. Indeed, we previously detected a direct interaction between pkp3 and F-actin (Gupta et al., 2023; doi: 10.3390/ijms24119458), although the precise actin-binding region of pkp3 remains to be identified. We also tested potential pkp4-actin interactions using the same in vitro binding assay (unpublished) but did not observe detectable binding. While these negative results do not exclude the possibility of a direct interaction, they indicate that further work will be required to elucidate this mechanism. It is worth noting that it took decades of research to clarify how the cadherin–catenin complex interacts with actin through α -catenin.

6). For a relation with desmosomes, this is only addressed by dsg2 staining in pkp 2/3 mutants. They should at least look at dsg2 colocalization with the cadherin complexes already in wt cells (Figure 1): how distributes the pkp4-cadherin clusters compared to desmosomes.

While the association of pkp4 with desmosomes has been extensively investigated previously (references provided), we nevertheless performed double staining of wild-type A431 cells for pkp4 and desmosomal marker desmoglein 2 (dsg2), as requested. Representative images and corresponding quantification are now included in the revised manuscript (Fig. S1c,d). Consistent with published reports, pkp4 and dsg2 are not co-localized, although lateral AJs and desmosomes are frequently observed in proximity.

Reviewer #3

We thank this reviewer for his notion that “The experimental data presented are convincing, and the identification of molecular constituents which mediate AJ functional diversity are a significant discovery which represent an important advance for the field.”

However, the reviewer considers that “the level of mechanistic insight presented (specifically, the lack of data demonstrating how these constituents exert their functional effects) is below that I would expect for an Article in JCB. It could be worth considering condensing the paper into a Report, a format where exciting discoveries with less mechanistic support are appropriate. I detail below the specific mechanistic / functional weaknesses, which admittedly would require a level of effort to address beyond the standard expectations of a revision, as well as several more minor issues.

Major mechanistic / functional weakness:

1) Mechanism of cytoskeleton coupling through pkp4

The authors convincingly show alpha-catenin independent coupling between pkp4 containing adhesions and the actin-myosin cytoskeleton, an exciting observation. However, they do not identify the alternative coupling mechanism. Doing so would substantially enhance the level of advance offered by the paper. This could be done e.g. by identifying additional pkp4-dependent constituents of these AJs and probing their role in cytoskeletal coupling.

We addressed a similar concern in our response to Reviewer #2 (see above). As noted, elucidating the molecular details of the interactions between lateral AJs and the cytoskeleton represents a separate and complex line of investigation, comparable to that required to understand α -catenin-dependent cadherin-actin interactions. Some of this work is already in progress in our laboratory; however, its completion will require considerable time.

2) Function of pkp4-containing lateral AJs

The authors also show that the pkp4-containing AJs that they identify are not necessary for cell adhesion by a standard assay, but the functional relevance of this class of adhesion is not further elucidated. The authors speculate that "dramatic effects observed in p120-KO cells were caused by the reduced level of E-cadherin but not by general inability of pkp4-CCC to maintain cell-cell adhesion" (lines 337-339). However, they do not provide any evidence for what these adhesions actually do. For example, do these lateral clusters interact with those in the neighboring cells, and do they play any role in the maturation of epithelial barriers? Adding such functional insights would also substantially strengthen the paper.

The potential role of lateral AJs in cell signaling is supported by their enrichment with PLEKHA5 and erbin. In addition, we have identified several other signaling proteins localized to these junctions, which we plan to report in the near future. Our current efforts are focused on elucidating the mechanisms that govern the recruitment of these proteins to lateral AJs. Understanding these mechanisms will allow us to generate cells in which these proteins are selectively mislocalized and to assess the resulting effects on specific cellular functions.

Our manuscript also cites experimental paper (line 86) suggesting a signaling role for lateral junctions. Furthermore, we included the requested experiment demonstrating that α -catenin-independent cadherin clusters are symmetrical across cell-cell contacts, consistent with their engagement in trans interactions (Fig. 7 c). This finding supports the idea that these junctions maintain close apposition of lateral membranes, which may be important for intercellular signaling events. Finally, we added a reference (line 352) showing that a lack of p120 affects cell-cell adhesion by reducing E-cadherin level.

Minor points:

4. The authors show that AJs can cluster along F-actin in the absence of a-catenin's actin binding region, but they do not show that pkp4 is required for this behavior. These AJ clusters are referred to as being generated by pkp4-CCC, but the authors don't show that pkp4 is present in the Triton X-100 resistant structures.

A new figure (Fig. 6 f) is added that shows pkp4 in the Triton X-100-resistant junctions.

5. The authors state that AJ clusters are formed by oligomerization of components inside the cell, but they do not assess how the extra-cellular regions of, e.g. E-cadherin, contribute to the clustering behavior of the different AJ classes.

This is a very interesting question, but it is not easy to answer conclusively. Expression of E-cadherin mutants lacking the trans-interaction interface exerts a dominant-negative effect that may indirectly disrupt lateral AJs. Indeed, we have examined this issue and found that such mutants cannot form lateral AJs even in α Cat-KO cells. However, because the underlying mechanism remains unclear, we consider these results too preliminary to include in the current manuscript.

6. The authors claim that "pkp4 appeared to be most prominent in AJs located at the middle portion of the lateral membrane" (lines 122-123), which is supported by a single image in Fig. 1c. This claim should be supported with quantification.

The ratio of fluorescence intensities of pkp4 to E-cadherin (pkp4/Ecad index) in different AJ has been added to the revised version (Fig. S1a).

Textual / presentation issues:

7. The paper would benefit from having the fluorescent images being presented in magenta / green, rather than red / green, for colorblind readers.

We apologize for not presenting our images in colors suitable for colorblind readers. We will make sure to follow this recommendation in the preparation of our next manuscript.

8. Figure 4 (b) legend states that "The arrows point at few remaining PLEKHA5-positive AJs in p120-KO cells.", but it appears that it is referring to pkp4-KO cells.

Thank you for pointing this out.

December 15, 2025

RE: JCB Manuscript #202507134R

Sergey Troyanovsky
Northwestern Medicine

Dear Dr. Troyanovsky,

Thank you for submitting your revised manuscript entitled "Two δ -Catenins, Plakophilin 4 and p120, Promote Formation of Distinct Types of Adherens Junctions." The manuscript has been re-assessed by two of the original reviewers. We would be happy to publish your paper in JCB pending final revisions necessary to meet our formatting guidelines (see details below).

A. MANUSCRIPT ORGANIZATION AND FORMATTING:

1) Text limits: Character count for Articles is < 40,000, not including spaces. Count includes title page, abstract, introduction, results, discussion, and acknowledgments. Count does not include materials and methods, figure legends, references, tables, or supplemental legends.

2) Figure formatting: Articles may have up to 10 main text figures. Scale bars must be present on all microscopy images, including inset magnifications. Molecular weight or nucleic acid size markers must be included on all gel electrophoresis. Please add scale bars to inset magnifications in figures S1C & S5 and MW markers to 4h.

Also, please avoid pairing red and green for images and graphs to ensure legibility for color-blind readers. If red and green are paired for images, please ensure that the particular red and green hues used in micrographs are distinctive with any of the colorblind types. If not, please modify colors accordingly or provide separate images of the individual channels.

3) Statistical analysis: Error bars on graphic representations of numerical data must be clearly described in the figure legend. The number of independent data points (n) represented in a graph must be indicated in the legend. Please indicate whether 'n' refers to technical or biological replicates (i.e. number of analyzed cells, samples or animals, number of independent experiments). If independent experiments with multiple biological replicates have been performed, we recommend using distribution-reproducibility SuperPlots (please see Lord et al., JCB 2020) to better display the distribution of the entire dataset, and report statistics (such as means, error bars, and P values) that address the reproducibility of the findings.

Statistical methods should be explained in full in the materials and methods. For figures presenting pooled data the statistical measure should be defined in the figure legends. Please also be sure to indicate the statistical tests used in each of your experiments (both in the figure legend itself and in a separate methods section) as well as the parameters of the test (for example, if you ran a t-test, please indicate if it was one- or two-sided, etc.). Also, if you used parametric tests, please indicate if the data distribution was tested for normality (and if so, how). If not, you must state something to the effect that "Data distribution was assumed to be normal but this was not formally tested."

4) Materials and methods: Should be comprehensive and not simply reference a previous publication for details on how an experiment was performed. Please provide full descriptions (at least in brief) in the text for readers who may not have access to referenced manuscripts. The text should not refer to methods "...as previously described." Please also briefly describe the SDS-PAGE methods and indicate the type of membrane used for immunoblotting.

5) For all cell lines, vectors, strains, constructs/cDNAs, etc. - all genetic material: please include database / vendor ID (e.g. Addgene, ATCC, etc.) or if unavailable, please briefly describe their basic genetic features, even if described in other published work or gifted to you by other investigators (and provide references where appropriate). Please be sure to provide the sequences for all of your oligos: primers, si/shRNA, RNAi, gRNAs, etc. in the materials and methods. You must also indicate in the methods the source, species, and catalog numbers/vendor identifiers (where appropriate) for all of your antibodies, including secondary. If antibodies are not commercial, please add a reference citation if possible.

6) Microscope image acquisition: The following information must be provided about the acquisition and processing of images:
a. Make and model of microscope
b. Type, magnification, and numerical aperture of the objective lenses

- c. Temperature
- d. Imaging medium
- e. Fluorochromes
- f. Camera make and model
- g. Acquisition software
- h. Any software used for image processing subsequent to data acquisition. Please include details and types of operations involved (e.g., type of deconvolution, 3D reconstitutions, surface or volume rendering, gamma adjustments, etc.).

7) References: There is no limit to the number of references cited in a manuscript. References should be cited parenthetically in the text by author and year of publication. Abbreviate the names of journals according to PubMed.

8) Supplemental materials: Articles may have up to 5 supplemental figures and 10 videos. Please also note that tables, like figures, should be provided as individual, editable files. A summary of all supplemental material should appear at the end of the Materials and methods section. Please include one brief sentence per item.

9) Video legends: Should describe what is being shown, the cell type or tissue being viewed (including relevant cell treatments, concentration and duration, or transfection), the imaging method (e.g., time-lapse epifluorescence microscopy), what each color represents, how often frames were collected, the frames/second display rate, and the number of any figure that has related video stills or images.

10) eTOC summary: A ~40-50 word summary that describes the context and significance of the findings for a general readership should be included on the title page. The statement should be written in the present tense and refer to the work in the third person. It should begin with "First author name(s) et al..." to match our preferred style.

11) Conflict of interest statement: JCB requires inclusion of a statement in the acknowledgements regarding competing financial interests. If no competing financial interests exist, please include the following statement: "The authors declare no competing financial interests." If competing interests are declared, please follow your statement of these competing interests with the following statement: "The authors declare no further competing financial interests."

12) A separate author contribution section is required following the Acknowledgments in all research manuscripts. All authors should be mentioned and designated by their first and middle initials and full surnames. We encourage use of the CRediT nomenclature (<https://casrai.org/credit/>).

13) ORCID IDs: ORCID IDs are unique identifiers allowing researchers to create a record of their various scholarly contributions in a single place. Please note that ORCID IDs are required for all authors. At resubmission of your final files, please be sure to provide your ORCID ID and those of all co-authors.

14) JCB requires authors to submit Source Data used to generate figures containing gels and Western blots with all revised manuscripts. This Source Data consists of fully uncropped and unprocessed images for each gel/blot displayed in the main and supplemental figures. For assays performed using capillary electrophoresis and/or immunoassay-based detection, authors should instead provide the electropherogram graph(s) for each experiment, plotting fluorescence/chemiluminescence intensity vs. molecular weight/size. Since your paper includes cropped gel and/or blot images, please be sure to provide one Source Data file for each figure gels, blots, and/or capillary electrophoresis assays along with your revised manuscript files. File names for Source Data figures should be alphanumeric without any spaces or special characters (i.e., SourceDataF#, where F# refers to the associated main figure number or SourceDataFS# for those associated with Supplementary figures). For traditional gels and blots, the lanes of the gels/blots should be labeled as they are in the associated figure, the place where cropping was applied should be marked (with a box), and molecular weight/size standards should be labeled wherever possible. For capillary electrophoresis assays, each trace in the graph should be color-coded and labeled to indicate which protein, gene, or sample is being measured (please try to avoid red/green combinations to accommodate our color-blind readers).

Source Data files will be directly linked to specific figures in the published article. Source Data Figures should be provided as individual PDF files (one file per figure). Authors should endeavor to retain a minimum resolution of 300 dpi or pixels per inch. Please review our instructions for export from Photoshop, Illustrator, and PowerPoint here: <https://rupress.org/jcb/pages/submission-guidelines#revised>.

15) Journal of Cell Biology now requires a data availability statement for all research article submissions. These statements will be published in the article directly above the Acknowledgments. The statement should address all data underlying the research presented in the manuscript. Please visit the JCB instructions for authors for guidelines and examples of statements at (<https://rupress.org/jcb/pages/editorial-policies#data-availability-statement>).

B. FINAL FILES:

Thank you for your attention to these final processing requirements. Please contact the journal office with any questions at cellbio@rockefeller.edu.

Thank you for this interesting contribution, we look forward to publishing your paper in Journal of Cell Biology.

Sincerely,

Ian Macara, PhD
Monitoring Editor
Journal of Cell Biology

Dan Simon, PhD
Scientific Editor
Journal of Cell Biology

Reviewer #2 (Comments to the Authors (Required)):

The authors considerably improved their manuscript with the required controls and complementary analysis. The reviewer thanks them for these efforts. Their answers on the amount of work needed for deciphering the underlying molecular mechanisms is convincing and the reviewer follow them in considering that the present message needs to be published rapidly to open the question of the mechanism of this novel junctional complex to a larger pool of researchers that could this way address it with their favourite approaches and thus accelerating discoveries.

Reviewer #3 (Comments to the Authors (Required)):

The authors have adequately responded to the issues raised in the previous round of review. While they have declined to pursue some of the more potentially impactful directions that were suggested due to the anticipated level of effort required, the revisions included have strengthened the manuscript. Notably, additional quantification has enhanced the rigor of the paper, and the new experiments included have strengthened the manuscript. I believe the paper is now overall acceptable for publication in JCB.